# DREAM-MPC: GRADIENT-BASED MODEL PREDICTIVE CONTROL WITH LATENT IMAGINATION

## ABSTRACT

State-of-the-art model-based Reinforcement Learning (RL) approaches either use gradient-free, population-based methods for planning, learned policy networks, or a combination of policy networks and planning. Hybrid approaches that combine Model Predictive Control (MPC) with a learned model and a policy prior to efficiently leverage the benefits of both paradigms have shown promising results. However, these hybrid approaches typically rely on gradient-free optimization methods, which can be computationally expensive for high-dimensional control tasks. While gradient-based methods are a promising approach, recent works have empirically shown that gradient-based methods often perform worse than their gradient-free counterparts due to the fact that gradient-based methods can converge to suboptimal local optima and are prone to exploding or vanishing gradients. We propose Dream-MPC, a novel approach that generates few candidate trajectories from a rolled-out policy and optimizes each trajectory by gradient ascent using a learned world model. We incorporate uncertainty regularization directly into the optimization objective and amortize optimization iterations over time by reusing previously optimized actions. We evaluate our method on multiple continuous control tasks from the DeepMind Control Suite, Meta-World and Humanoid-Bench and show that gradient-based MPC can significantly improve the performance of the underlying policy and can outperform gradient-free MPC and state-of-the-art baselines. To facilitate further research on gradient-based MPC, we will open source our code and more at `https://dream-mpc.github.io`.

## 1 INTRODUCTION

Reinforcement Learning has achieved promising results in recent years and demonstrated its application to robotics (Wu et al., 2023; Lancaster et al., 2024; Seo et al., 2025). However, model-free methods often struggle with sample efficiency and generalization, especially in complex and high-dimensional environments (Byravan et al., 2022). Model-based RL, on the other hand, can be more sample-efficient and can generalize better, but requires an accurate model of the environment (Xiao et al., 2019). There has been growing interest in world models that are learned from data and can be used for decision-making (Sutton, 1991; Ha & Schmidhuber, 2018). Many recent works (Hafner et al., 2019; Hansen et al., 2022; 2024; Srinivas et al., 2018) use a learned world model for planning through imaginary rollouts with Model Predictive Control (MPC) (Richalet et al., 1978; Cutler & Ramaker, 1979) and rely on gradient-free, sampling-based methods such as the Cross Entropy Method (CEM) (Rubinstein, 1997) or Model Predictive Path Integral (MPPI) (Williams et al., 2015; 2017) for trajectory optimization. Although sampling-based MPC methods can be parallelized using Graphics Processing Units (GPUs), their implementation on embedded systems can be challenging due to the limited computational resources. In addition, planning with sampling-based methods is highly inefficient or even intractable in high-dimensional spaces, which might limit their applicability to real-world robotics tasks (Xie et al., 2021).

In contrast, fully amortized methods such as Dreamer (Hafner et al., 2020) learn a purely reactive policy via imaginary rollouts. Inference for the learned policy is computationally less expensive than the search procedure using CEM. However, amortized policies often lack generalization (Byravan et al., 2022). Since the learned world models are usually differentiable, it is quite natural to propose gradient-based methods for trajectory optimization because they can be more efficient than gradient-free, sampling-based methods. Instead of sampling many action sequences and evaluating them as

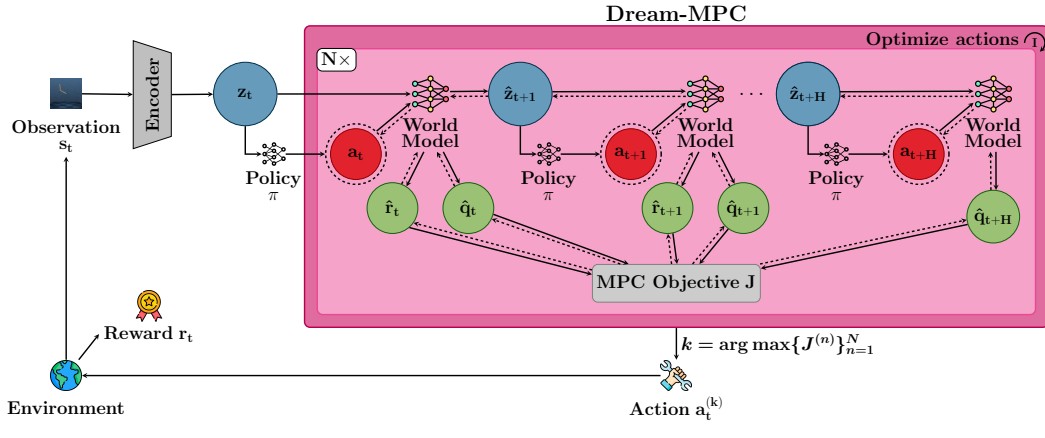

Figure 1: **Overview of the proposed approach.** Dream-MPC optimizes action sequences rolled out from a policy network $\pi$ in latent space $z$ with gradient-based MPC. $N$ candidate trajectories are sampled from the policy prior and optimized for $I$ iterations using gradient ascent to maximize the objective $J$. The first action with the highest predicted return is applied, and the procedure is repeated for the next time step. The policy network and world model are shared across candidates and time steps.

done by CEM, gradients backpropagated through the model can be used to guide the optimization procedure (Bharadhwaj et al., 2020). When the action dimension increases, there is an exponential growth in search space for CEM, while there is only a small increase in computational load for gradient descent, i.e., an additional gradient dimension (Bharadhwaj et al., 2020). While few works propose to combine gradient-based optimization with world models, the empirical results observed were worse than for their gradient-free counterparts (Bharadhwaj et al., 2020; S V et al., 2023; Zhou et al., 2025).

We propose Dream-MPC, a novel method which combines gradient-based MPC with a learned policy network and world model. Our method incorporates uncertainty directly into the optimization objective and amortizes optimization iterations over time to further improve performance and computational efficiency. We evaluate our method empirically on various tasks from different domains, including high-dimensional tasks and tasks with visual observations, as well as for different model-based RL algorithms with distinct types of world models and when using gradient-based MPC during training. The results show that our method can significantly improve the performance of the policy and even outperform its gradient-free equivalent and state-of-the-art methods.

## 2 RELATED WORK

**Model-based RL.** Model-based RL tries to learn a model of the environment that can be used to predict the outcome of actions and plan accordingly (Sutton, 1991). World models are considered a central component of human thinking and decision-making processes (Sutton, 1991; Ha & Schmidhuber, 2018; LeCun, 2022). While some approaches to world modelling show promising results and are able to generalize to different domains, they are mostly focused on representation learning and not or only partially cover the planning aspect. The combination of elements of planning and search (especially Monte Carlo Tree Search) with deep reinforcement learning has shown remarkable successes in game domains (Silver et al., 2016; 2017a). Most recent model-based RL approaches use the learned world model for planning through imaginary rollouts (Srinivas et al., 2018; Micheli et al., 2023; Hansen et al., 2024; Hafner et al., 2025; Mosbach et al., 2025). However, the performance of these approaches depends heavily on the quality of the learned world model (Talvitie, 2014) and often suffers from the compounding error problem (Asadi et al., 2019).

**MPC and RL.** State-of-the-art approaches such as those from the Dreamer family (Hafner et al., 2020; 2021; 2025) use a policy network to predict the actions directly. While policy networks have

shown remarkable success for robotics applications, the world model and value function are typically only utilized during training, and the policy is then frozen during inference. This procedure leads to a reactive policy, which can be considered as offline planning and limits the generalization capabilities (Byravan et al., 2022). To address this limitation, recent works such as TD-MPC (Hansen et al., 2022; 2024), POLO (Lowrey et al., 2019) or PlaNet (Hafner et al., 2019) combine model-based RL with online planning through MPC to leverage the benefits of both paradigms. Typically, MPC is performed using gradient-free, sampling-based methods such as CEM or MPPI. Although, the results obtained empirically are often good, for each time step, hundreds or thousands of different action alternatives are sampled and evaluated, which increases the computational effort and renders these approaches only partly suitable for real-time applications.

**Gradient-based Planning.** The idea of gradient-based planning has been around for decades (Kelley, 1960) and typically refers to backpropagating gradients of a cost or reward function with respect to actions to iteratively optimize a sequence of actions by gradient descent. While early works relied on known analytic forms of environment dynamics, more recent works revisited the idea with learned approximate models of the environment (Srinivas et al., 2018; Silver et al., 2017b; Henaff et al., 2018). However, there are only a few works that have been able to successfully perform gradient-based planning and these approaches are usually limited since they either require expert demonstrations (Srinivas et al., 2018) or cannot scale to more challenging robotics tasks (Henaff et al., 2018). Works such as (Bharadhwaj et al., 2020) and (S V et al., 2023) use a Gaussian as a proposal distribution for gradient-based optimization. Typically, a more informative proposal is used for MPC to warm-start the optimization procedure, for example a policy network. Prior works which combine policy models and MPC mostly use the policy model to generate a trajectory which is then optimized using gradient-free methods (Byravan et al., 2022; Mansard et al., 2018; Hamrick et al., 2021; Argenson & Dulac-Arnold, 2021; Morgan et al., 2021). Since the learned world models are usually differentiable, also gradient-based methods have been proposed for optimizing the trajectory proposal from a policy model (S V et al., 2023). However, gradient-based optimization methods perform worse in their experiments compared to their gradient-free counterparts. The reasons are attributed to problems with the gradients, but are not analyzed in detail.

Note that while the general idea of combining policy networks with MPC itself is not new, previously proposed methods have only been applied to few and relatively simple tasks without systematically evaluating their performance. To the best of our knowledge, we are the first to achieve a gradient-based MPC method with a learned world model that can outperform its gradient-free equivalent and state-of-the-art baselines by introducing uncertainty regularization and reusing previously planned actions. We have also evaluated the performance of gradient-based MPC for a broad variety of environments, including state- and image-based observations and different types of world models. We provide a summary over the main differences between Dream-MPC and hybrid Grad-MPC (S V et al., 2023) in Appendix E.

## 3 PRELIMINARIES

**Reinforcement Learning** can be formulated as an infinite-horizon Markov Decision Process (MDP) with continuous action and state spaces, which can be defined as a tuple $\langle \mathcal{S}, \mathcal{A}, \mathcal{T}, \mathcal{R}, \gamma \rangle$, where $\mathcal{S}$ and $\mathcal{A}$ are the state and action spaces, $\mathcal{T} : \mathcal{S} \times \mathcal{A} \to \mathcal{S}$ is the transition or dynamics function, $\mathcal{R} : \mathcal{S} \times \mathcal{A} \to \mathbb{R}$ is the reward function and $\gamma$ is a discount factor. The goal is to obtain a policy $\pi : \mathcal{S} \to \mathcal{A}$, which maximizes the expected discounted sum of rewards, i.e., the return $\mathbb{E}_\pi[\sum_{t=0}^{\infty} \gamma^t r_t]$, where $r_t = \mathcal{R}(s_t, \pi(s_t))$. Model-based RL learns a model of the environment, often referred to as world model, which is then used for selecting actions and deriving a policy by planning with the learned model.

**Model Predictive Control** is a well-known method for trajectory optimization, which minimizes a cost function over a finite horizon while taking the system dynamics and constraints into account. The optimization problem is solved at each time step, using the current state as initial condition and the predicted future states. The solution provides the optimal action sequence for the next few time steps with respect to the predicted costs. Thus, MPC generates a locally optimal sequence of actions up to the prediction horizon $H$ by solving the following optimization problem:

$$\pi(s_t) = \arg\max_{a_{t:t+H}} \mathbb{E}\left[\sum_{i=0}^{H} \gamma^{t+i} \mathcal{R}(s_{t+i}, a_{t+i})\right]. \tag{1}$$

The learned model is used to estimate the return of a candidate trajectory (Negenborn et al., 2005). Since solving Eq. (1) leads to a locally optimal solution and is not guaranteed to solve the general RL problem outlined before, most state-of-the-art methods learn value functions to bootstrap return estimates beyond the horizon $H$.

## 4 DREAM-MPC: GRADIENT-BASED MODEL PREDICTIVE CONTROL

We propose Dream-MPC, which uses gradient ascent to optimize action sequences sampled from a policy network in an MPC-like manner. The idea is shown in Fig. 1. Since gradient ascent is prone to getting stuck at local optima, we propose to generate few candidate trajectories by sampling from a stochastic policy network. Instead of sampling thousands of trajectories from a Gaussian distribution like CEM, we only consider few trajectories based on the policy. Namely, for each time step $t$, the algorithm creates $N$ initial action sequences by performing an imaginary rollout of a stochastic policy $\pi_\theta$ in latent space $z$ using a learned latent dynamics model $d$:

$$\hat{a}_\tau^{(n)} \sim \pi_\theta(\cdot|z_\tau^{(n)}), \quad z_{\tau+1}^{(n)} = d(z_\tau^{(n)}, \hat{a}_\tau^{(n)}), \quad \text{with} \quad \tau = t, ..., t+H, \quad n = 1, ..., N. \quad (2)$$

In case of a deterministic policy we add small perturbations to the initial action sequence sampled from the policy to generate $N$ candidate trajectories. The learned world model predicts the following latent states as well as the rewards $\hat{r}$ for each state and the terminal values $\hat{q}$. Each trajectory is then refined using gradient ascent with step size $\alpha$ to maximize the respective expected return, which is estimated using the predictions from the world model. The first action of the candidate trajectory with the highest expected return is applied, and the planning procedure is repeated in the next time step. Sampling from a policy provides a warm-start through proposing a decent initial solution for the optimization, which has been shown to be essential for the performance of gradient-free (Hansen et al., 2022) and gradient-based optimization methods (Parmas et al., 2018). Our method allows for combining the benefits of both, fully amortized methods using reactive policies and fully online planning, namely improved generalization while reducing computational costs. In contrast to naively sampling random action sequences, which do not leverage any knowledge of the optimization problem, our approach uses gradients backpropagated through the learned world model to efficiently guide the optimization.

Since we optimize actions over a receding horizon, but only apply the first action at each time step, we propose to amortize optimization iterations over time by reusing corresponding optimized actions from previous time steps to initialize actions as a mixture of previously optimized action $\tilde{a}$ and policy actions $\hat{a}$:

$$a_\tau^{(n)} = \rho \cdot \tilde{a}_{\tau-1}^{(n)} + (1 - \rho) \cdot \hat{a}_\tau^{(n)}, \quad n = 1, ..., N, \quad (3)$$

where $\rho$ is the reuse coefficient, which controls the influence of the previously optimized actions. For the action at time step $t + H$, there is no previously planned action. Thus, we initialize the planned action by the same value as the planned action of the time step before.

For our experiments, we integrate our method into TD-MPC2 (Hansen et al., 2024), a model-based RL algorithm, which performs local trajectory optimization using MPPI in the latent space of a learned world model. Instead of learning a dynamics model using a reconstruction objective, TD-MPC2 implicitly learns a control-centric world model from environment interactions using a combination of joint-embedding prediction, reward prediction, and TD-learning without decoding observations.

The TD-MPC2 architecture consists of following five learned components:

| | | |
|---|---|---|
| Encoder | $z_t = h(s_t)$ | (maps observations to latent representations), |
| Latent dynamics | $z_{t+1} = d(z_t, a_t)$ | (predicts latent forward dynamics), |
| Reward | $\hat{r}_t = R(z_t, a_t)$ | (predicts reward $r$ of a transition), |
| Terminal value | $\hat{q}_t = Q(z_t, a_t)$ | (predicts discounted sum of rewards, i.e., return), |
| Policy prior | $\hat{a}_t \sim \pi_\theta(z_t)$ | (predicts action $a^*$ that maximizes $Q$), |

where $s$ and $a$ are the states and actions, and $z$ is the latent representation. Since we only consider single-task experiments in this work, we omit the learnable task embedding used for multi-task world models.

The policy prior $\pi_\theta$ serves to guide the sampling-based MPPI trajectory optimizer in TD-MPC2 as well as our gradient-based method. TD-MPC2 maintains a replay buffer $\mathcal{B}$ during online interaction, which is used to iteratively update the world model and collect new environment data by planning with the learned model. Please refer to Appendix B for details on the model training, architecture and MPPI planning procedure. We replace the MPPI planner by our gradient-based MPC method.

---

**Algorithm 1: Dream-MPC**

---

**Input:** Encoder $h(s)$, dynamics model $d(z, a)$, reward model $R(z, a)$, value function model $Q(z, a)$, policy prior $\pi_\theta(z)$, current state $s_t$, planning horizon $H$, optimization iterations $I$, candidates per iteration $N$, action optimization rate $\alpha$

Encode state into latent representation $z_t \leftarrow h(s_t)$.
Sample $N$ action sequences by rolling out the policy $\pi_\theta$ with the latent dynamics model $d$.
Initialize candidate action sequences $a_{t:t+H}$ via Eq. (3).
**for** *optimization iteration* $i = 1, 2, \ldots I$ **do**
   **for** *candidate action sequence* $n = 1, 2, \ldots N$ **do**
      **for** *rollout step* $\tau = t \ldots t + H - 1$ **do**
         Predict reward $\hat{r}_\tau^{(n)} = R(z_\tau, a_\tau)$.
         Predict uncertainty $u_\tau^{(n)}$ via Eq. (5).
         Predict next latent state $z_{\tau+1}^{(n)} \leftarrow d(z_\tau, a_\tau)$.
      Predict terminal value $\hat{q}_{t+H}^{(n)} = Q(z_{t+H}, a_{t+H})$.
      Compute optimization objective $J^{(n)}$ using $\hat{r}$, $\hat{q}$ and $u$ via Eq. (6).
      Optimize action sequence via $a_{t:t+H}^{(n)} \leftarrow a_{t:t+H} + \alpha \nabla_a J^{(n)}$.

**Output:** First optimized action $a_t^{(k)}$ with $k = \arg\max_n \{J^{(n)}\}_{n=1}^N$.

---

Our gradient-based MPC algorithm is summarized in Alg. 1. The MPC procedure requires $N \times I \times H$ evaluations of the world model at each time step, which equals $512 \times 6 \times 3 = 9216$ for MPPI while our method uses significantly less model evaluations, i.e., only $5 \times 1 \times 3 = 15$. Note that while we use TD-MPC2 for our experiments, our method can also be integrated into other model-based reinforcement learning approaches such as Dreamer (Hafner et al., 2020) or DINO-WM (Zhou et al., 2025). We include results and implementation details on integrating our method into Dreamer in Appendix D.

We further integrate our method into BMPC (Wang et al., 2025), which builds on TD-MPC2 and learns a policy $\pi_\theta$ by imitating an MPC expert $\pi_{\text{MPC}}$ and at the same time uses the policy to guide the MPC optimization process. Thus, the policy is learned using the following objective:

$$\mathcal{L}_\pi(\theta) \doteq \underset{(\mathbf{s},\mathbf{a})_{0:H} \sim \mathcal{B}}{\mathbb{E}} \left[ \sum_{t=0}^{H} \lambda^t \left[ \text{KL}(\pi_{\text{MPC}}(\cdot | h(\mathbf{s}_t), \pi_\theta), \pi_\theta(\cdot | \mathbf{z}_t)) / \max(1, S) - \beta \mathcal{H}(\pi_\theta(\cdot | \mathbf{z}_t)) \right] \right],$$

$$\mathbf{z}_0 = h(\mathbf{s}_0), \ \mathbf{z}_{t+1} = d(\mathbf{z}_t, \mathbf{a}_t),$$

$$S \doteq \text{EMA}(\text{Per}(\text{KL}(\pi_{\text{MPC}}, \pi_\theta), 95) - \text{Per}(\text{KL}(\pi_{\text{MPC}}, \pi_\theta), 5), 0.99),$$

(4)

where $\mathcal{H}$ is the entropy, KL is the Kullback-Leibler divergence, $\mathbf{z}_{0:H}$ are latent vectors rolled out using the models $h$ and $d$, and $\beta$ and $\lambda$ are hyperparameters for loss balancing and temporal weighting, respectively. The KL loss is normalized using moving percentiles $S$, which are commonly used to stabilize training. The results of Wang et al. (2025) show that this bootstrapping approach can improve sample efficiency and asymptotic performance, especially for high-dimensional tasks. We use BMPC since it provides a higher quality policy compared to TD-MPC2, where the performance gap between the policy network and the MPC procedure is quite large as shown in Appendix C.3. For more details on BMPC, please refer to Appendix B.2.

We further propose to regularize the planning procedure by penalizing trajectories with a large uncertainty because our method may benefit from conservative value estimations given that the estimates are directly used for optimizing the actions. Therefore, we estimate the (epistemic) uncertainty of a trajectory as proposed by Hansen et al. (2024) for offline RL and multi-task world models:

$$u_t = \text{avg}([\hat{q}_1, \hat{q}_2, \ldots, \hat{q}_M]) \cdot \text{std}([\hat{q}_1, \hat{q}_2, \ldots, \hat{q}_M]),$$

(5)

where $\hat{q}_m$ is the predicted value from Q-function $m$ from an ensemble of $M$ Q-functions. The regularization strength at each time step is scaled based on the magnitude of the mean value predictions

for a given latent state to account for different tasks without requiring task-specific coefficients. The planning objective is then redefined as:

$$J = \sum_{h=t}^{H-1} \left( \gamma^h \cdot R(z_h, a_h) - \lambda_{\text{unc}} \cdot u_h \right) + \gamma^H \cdot Q(z_{t+H}, a_{t+H}) - \lambda_{\text{unc}} \cdot u_{t+H}, \quad (6)$$

where $\lambda_{\text{unc}}$ is a task-agnostic coefficient that balances return maximization and uncertainty minimization. While this requires to specify a coefficient that weighs both aspects, we found it sufficient in our experiments to set $\lambda_{\text{unc}} = 0.01$. All hyperparameters specific to Dream-MPC are listed in Tab. 1. We also conduct experiments in which we use this uncertainty regularization for TD-MPC2 and BMPC and include the results in Appendix C.3.

Table 1: **Dream-MPC Hyperparameters.** We use the same hyperparameters for all tasks. All other hyperparameters are the default TD-MPC2 and BMPC values respectively.

| Hyperparameter | Value |
|---|---|
| **Planning** | |
| Iterations $I$ | 1 |
| Policy prior samples $N$ | 5 |
| Optimization step size $\alpha$ | 0.1 |
| Action reuse coefficient $\rho$ | 0.1 |
| Uncertainty regularization coefficient $\lambda_{\text{unc}}$ | 0.01 |

## 5 EXPERIMENTS

We evaluate our method on a set of 24 diverse continuous control tasks from the DeepMind Control Suite (Tassa et al., 2020), HumanoidBench (Sferrazza et al., 2024) and Meta-World (Yu et al., 2019) covering a wide range of task difficulties including high-dimensional state and action spaces, sparse rewards, complex locomotion, and manipulation. Additionally, we also include results for six DM-Control tasks with visual observations. For details on the environments, please refer to Appendix A.

### 5.1 COMPARISON TO BASELINES

We compare our method to following state-of-the-art baselines commonly used for continuous control tasks:

- Soft-Actor-Critic (SAC) (Haarnoja et al., 2018), a model-free RL method which uses a maximum entropy objective for policy learning,
- Dreamer-v3 (Hafner et al., 2025), a model-based RL method which learns a policy network using rollouts from a generative world model,
- TD-MPC2 (Hansen et al., 2024), a model-based RL method which uses policy-guided MPPI for action selection, and
- BMPC (Wang et al., 2025), an extension of TD-MPC2 which uses imitation learning of the MPC planner for policy learning.

We first evaluate the performance of Dream-MPC using (pre-)trained TD-MPC2 and BMPC models, respectively, when replacing the MPPI planner by our proposed gradient-based MPC planner at test time. For TD-MPC2, we use the models provided by Hansen et al. (2024) for the DeepMind Control Suite and Meta-World, except for Cartpole Swingup Sparse, Dog Run, Dog Walk, Humanoid Run and Humanoid Walk because some checkpoints cannot be loaded after code restructuring[1]. Thus, we trained new models for these tasks as well as for HumanoidBench. We further train BMPC, Dreamer-v3 and SAC models for all tasks. For more details on the baselines refer to Appendix B.

We report performance metrics across all 24 tasks using the *rliable*[2] package provided by Agarwal et al. (2021) to evaluate the performance of our method. Specifically, we report the optimality gap,

---

[1]cf. https://github.com/nicklashansen/tdmpc2/issues/23
[2]https://github.com/google-research/rliable

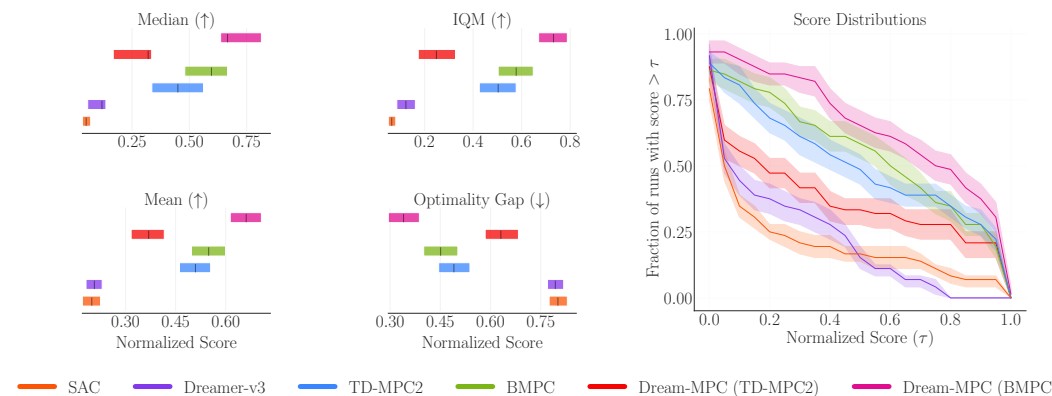

Figure 2: **Aggregate performance metrics.** Left: optimality gap, interquartile median (IQM), mean and median normalized scores with 95% confidence intervals. Right: score distributions across all tasks, which provides insights into the variance of the performance. Notably, Dream-MPC achieves the best results. Detailed results are included in Tabs. 9 to 11.

median, interquartile median (IQM), and mean normalized scores as well as the performance profile curves with 95% confidence intervals based on the evaluation scores of trained BMPC agents in Fig. 2. Confidence intervals are estimated using the percentile bootstrap with stratified sampling as recommended by Agarwal et al. (2021). For a comparison across different score scales of all tasks, we normalize DMControl scores by diving by 1000, and HumanoidBench scores as proposed in Lee et al. (2025):

$$\text{Normalized-Score}(x) = \frac{x - \text{random score}}{\text{target score} - \text{random score}}, \tag{7}$$

where we use the random and target success scores provided by the authors. Please refer to Lee et al. (2025) for more details. Meta-World scores are left as they are since the success rates are already values between zero and one. The detailed evaluation results for all environments are shown in Tabs. 9 to 11. Our gradient-based MPC method can improve the performance of the policy network and outperforms MPPI when using BMPC as a basis. While Dream-MPC can also significantly improve the performance of the underlying policy for TD-MPC2, it cannot consistently match the performance of MPPI because for TD-MPC2 there is a relatively large gap between the performance of the policy only and with MPPI as shown in Appendix C.3. This highlights the need for a good policy proposal for gradient-based MPC, especially for high-dimensional problems. We discuss this in more detail in Appendix C.2.

Additionally, we evaluate the performance of our method using image-based observations to demonstrate that our method also works well in these settings. The results are shown in Tab. 2. We find that our method can also improve the performance of the underlying policy and even outperforms MPPI for visual observations.

Table 2: **Visual observations.** Performance comparison of different BMPC variants on tasks from the DeepMind Control Suite using image-based observations.

| Environment | BMPC | BMPC (policy only) | Dream-MPC (BMPC) |
|---|---|---|---|
| Acrobot Swingup | $287 \pm 45$ | $\mathbf{292 \pm 18}$ | $\underline{288 \pm 31}$ |
| Cartpole Swingup Sparse | $\underline{709 \pm 120}$ | $625 \pm 283$ | $\mathbf{725 \pm 141}$ |
| Cheetah Run | $\underline{609 \pm 23}$ | $597 \pm 45$ | $\mathbf{643 \pm 9}$ |
| Hopper Hop | $253 \pm 11$ | $\underline{264 \pm 6}$ | $\mathbf{275 \pm 3}$ |
| Quadruped Walk | $\underline{427 \pm 78}$ | $402 \pm 44$ | $\mathbf{435 \pm 76}$ |
| Walker Run | $740 \pm 15$ | $\underline{740 \pm 6}$ | $\mathbf{762 \pm 6}$ |

The results are the mean episode returns and standard deviations for three random seeds and ten test episodes. **Best** and second best results are highlighted.

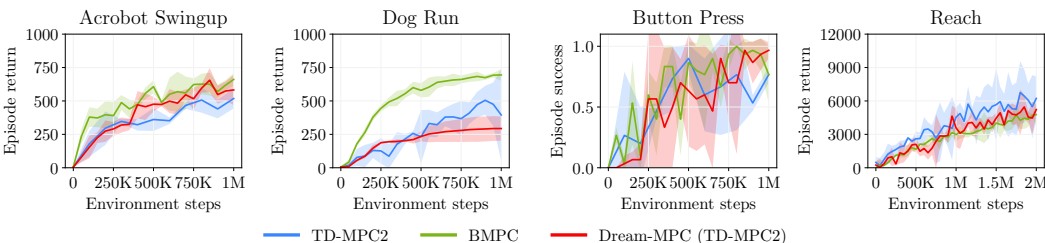

Figure 3: **Learning curves for four tasks from the DeepMind Control Suite.** The line represents the mean episodic return and the shaded area the 95% confidence interval across 3 seeds.

In addition to analyzing our gradient-based MPC method only during inference, we also evaluate its performance when it is already being used during training. Therefore, we use TD-MPC2 as a basis without imitation learning because we hypothesize that the bootstrapping approach of BMPC might lead to unstable training and premature convergence, especially since we have only few candidate trajectories. While combining gradient-based MPC with imitation learning is an interesting research direction, we leave this for future work. Fig. 3 shows the learning curves of BMPC, TD-MPC2 and of Dream-MPC for four different environments. Overall, our gradient-based MPC planner can match the performance of TD-MPC2's MPPI planner. While for simpler control problems Dream-MPC can even outperform TD-MPC2 and match BMPC, we find that for high-dimensional problems our method performs slightly worse. This issue may result from premature convergence due to less diversity among the few candidate trajectories compared to MPPI. We also find improvements in sample-efficiency and asymptotic performance when integrating our method into Dreamer. The results are shown in Appendix D.1.

We benchmark inference times of the different methods on a single Nvidia GeForce RTX 4090 GPU. The results in Tab. 3 show that Dream-MPC is about as fast as MPPI for lower dimensional problems, potentially enabling its usage for real-world robotics applications, which require high control frequencies. While there is an increase in inference time for high-dimensional problems, our method is still significantly faster as for example Grad-MPC (S V et al., 2023), which samples hundreds of action sequences from a Gaussian and optimizes each candidate solution for multiple iterations by using gradient ascent. The corresponding inference times are shown in Tab. 16.

Table 3: **Inference times of different methods for Acrobot Swingup.** Mean and standard deviation for three random seeds and ten test episodes per seed.

| Method | Inference time | Method | Inference time |
|---|---|---|---|
| BMPC | $18.77 \pm 0.11$ ms | TD-MPC2 | $20.83 \pm 0.14$ ms |
| Dream-MPC (BMPC) | **$18.15 \pm 0.12$ ms** | Dream-MPC (TD-MPC2) | **$19.53 \pm 0.11$ ms** |

## 5.2 ABLATION STUDY

We perform ablations to evaluate our design choices and provide insights into which components are crucial to successfully perform gradient-based MPC. Using a high-quality policy prior to warm-start the MPC optimization is particularly important for high-dimensional problems, as shown in Tab. 4. Together with reusing previously optimized actions, warm-starting reduces computational costs. We replace the policy prior by a Gaussian distribution to highlight the importance of a good initial proposal distribution to warm-start the MPC process and use the same number of candidate trajectories as MPPI, i.e., 512. For a fair comparison, we compensate for the less informative prior by increasing the number of optimization iterations to five, which, depending on the environment, leads to an increase in inference time by a factor of about five to ten compared to Dream-MPC. We further find that uncertainty regularization and amortization of optimization iterations by reuse of previous planned actions are especially important when using gradient-based MPC during training, as illustrated in Fig. 4a. Fig. 4b shows a sensitivity analysis of the uncertainty regularization and reuse coefficients, emphasizing that Dream-MPC is quite robust to the choice of these parameters. We also conduct experiments in which we use this uncertainty regularization for TD-MPC2 and

BMPC and include the results in Appendix C.3. The results indicate that for BMPC, the performance slightly improves – except for HumanoidBench – while for TD-MPC2, the uncertainty regularization leads to a performance decrease for all three domains. Additionally, we provide an analysis of the planner gradients when integrating our method into Dreamer in Appendix D.2, which suggests that Dream-MPC is more robust, compared to Grad-MPC.

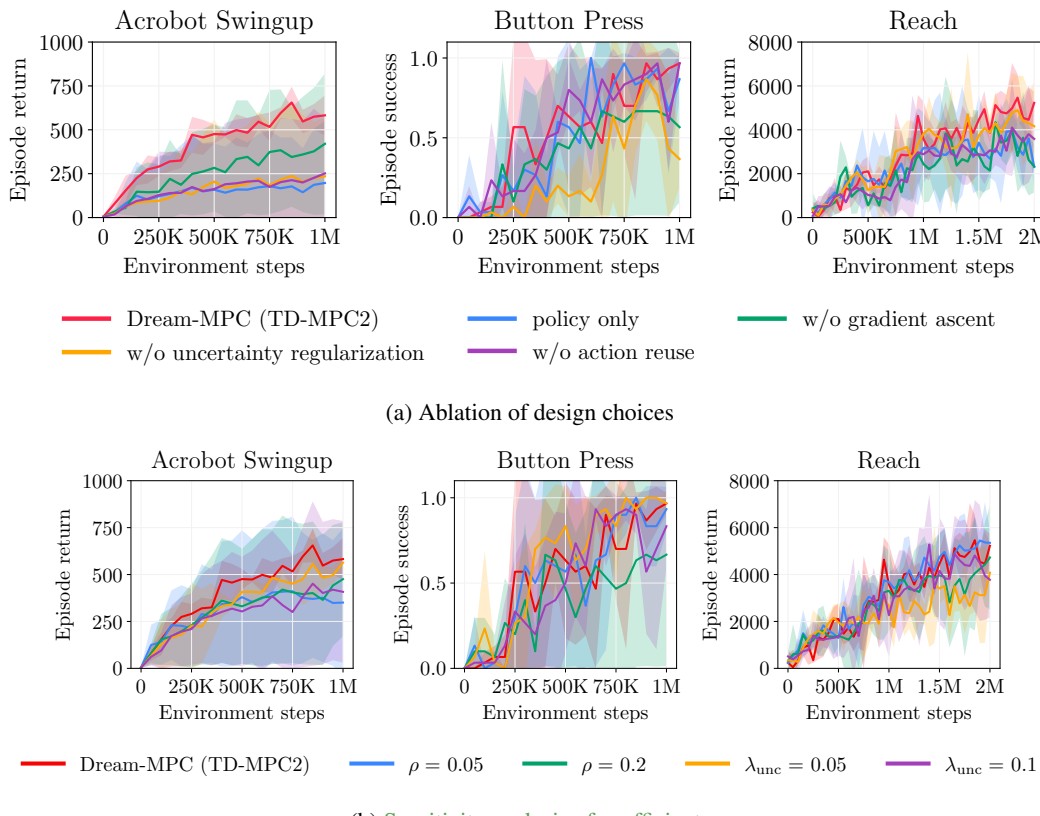

(a) Ablation of design choices

(b) Sensitivity analysis of coefficients

Figure 4: **Ablations.** (a) Performance of different Dream-MPC (TD-MPC2) variants demonstrating the importance of each design choice. (b) Performance of Dream-MPC (TD-MPC2) with different uncertainty regularization and action reuse coefficients. The line represents the mean episodic return and the shaded area the 95% confidence interval across 3 seeds.

Table 4: **Dream-MPC ablations.** We compare the performance of different variants using trained BMPC models.

| Method | Acrobot Swingup | Humanoid Run | Button Press | Reach |
|---|---|---|---|---|
| Dream-MPC (BMPC) | **596 ± 50** | **531 ± 38** | 0.67 ± 0.47 | **4348 ± 215** |
| w/o MPC (policy-only) | 564 ± 52 | 458 ± 15 | **1.0 ± 0.0** | 2117 ± 309 |
| w/o policy prior | 554 ± 21 | 7 ± 4 | 0.70 ± 0.22 | 842 ± 239 |
| w/o gradient ascent | 579 ± 43 | 496 ± 25 | 0.97 ± 0.05 | 2362 ± 323 |

The results are the mean episode returns and standard deviations for three random seeds and ten test episodes. **Best** and second best results are highlighted.

We further evaluate the performance of fully trained BMPC agents with gradient-based MPC when varying the number of candidates, the number of optimization iterations, and the planning horizon. The results for Acrobot Swingup, Humanoid Run and Slide are shown in Fig. 5. All other hyperparameters are fixed to their default value when varying one. While we use a single set of hyperparameters across all environments, algorithms, and for state-based and visual observations, we find that dynamically adjusting the planning parameters can help to further improve performance. The parameter sweep also shows that increasing the horizon and the number of optimization iterations

does not necessarily always increase the performance further, but can also impair the performance for some environments. This issue may result from an inaccurate model, especially when using a longer prediction horizon than the one used for training the model.

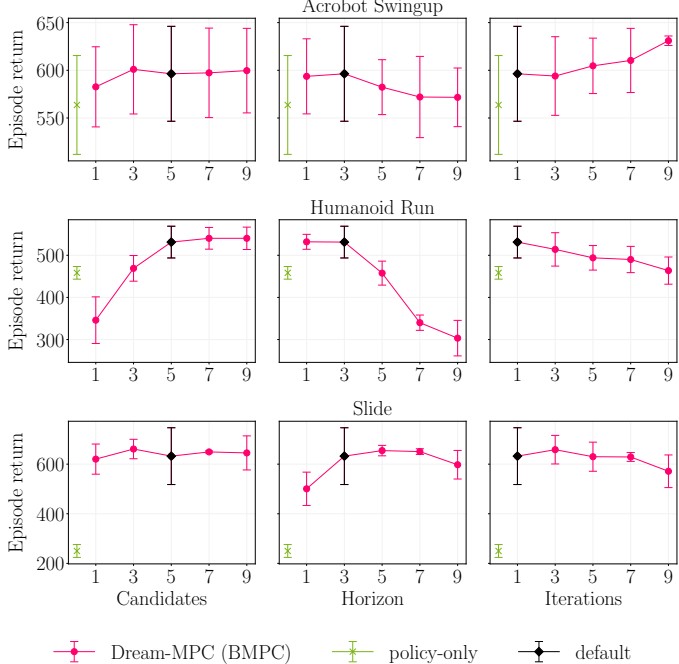

Figure 5: **Parameter sweep.** Performance of trained BMPC agents with Dream-MPC at test time when varying the number of candidates, horizon and number of optimization iterations. When varying one hyperparameter, the others are fixed to their default value. We also include the performance of the learned policy $\pi_\theta$ and the default values of one iteration, a horizon of three and five candidate trajectories.

## 6 CONCLUSION

We propose Dream-MPC, a novel method for gradient-based planning with a learned policy network and world model, which incorporates amortization of optimization iterations over time and uncertainty to overcome the limitations of previously proposed gradient-based MPC methods, namely worse performance compared to their gradient-free equivalents and high computational costs. We evaluate our method on a broad set of diverse tasks from different domains, including visual observations, to demonstrate its effectiveness. Our empirical evaluation shows that Dream-MPC can not only outperform the baselines, but is also more robust to hyperparameters and faster compared to previously proposed gradient-based MPC methods. Overall, our results highlight that gradient-based trajectory optimization with a learned world model has the potential to significantly improve the performance of model-based RL algorithms.

Our experiments suggest that it may be beneficial to dynamically adapt the optimization parameters such as the action optimization step size and number of iterations to further improve the performance, especially for high-dimensional problems. As our current approach is applied to single-task problems, it would also be interesting to extend it to multi-task world models to evaluate its potential in this setting.

### REPRODUCIBILITY STATEMENT

To ensure reproducibility of our work and encourage further research on gradient-based MPC, we have included details including hyperparameters of our proposed method as well as for the baselines in Section 4 and Appendix B. We will also release our source code and more at `https://dream-mpc.github.io`.

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

## A  ENVIRONMENT DETAILS

We evaluate our method on a total of 24 continuous control tasks from three different domains: eight environments from the Deep Mind Control suite, including four high-dimensional locomotion tasks, eight environments from HumanoidBench, and eight environments from Meta-World. All three domains are infinite-horizon continuous control environments for which we use a fixed episode length, an action repeat of 2 for the DeepMind Control Suite and Meta-World and 1 for HumanoidBench, and no termination conditions. We follow the success definition of Hansen et al. (2024). This section provides an overview and details for all tasks considered, including their observation and action dimensions.

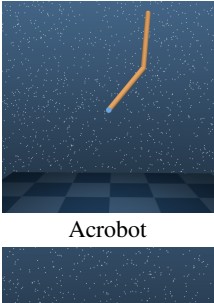 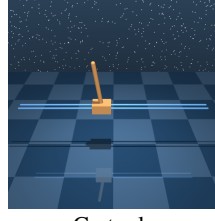 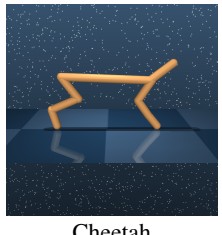 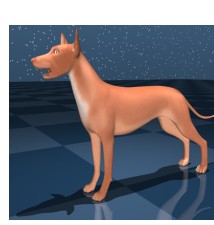

| Acrobot | Cartpole | Cheetah | Dog |

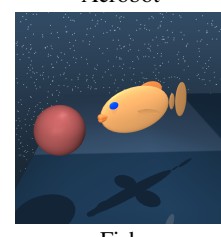 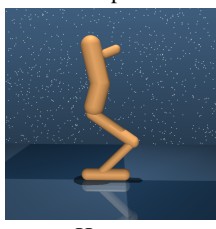 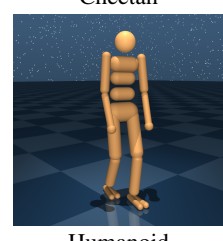 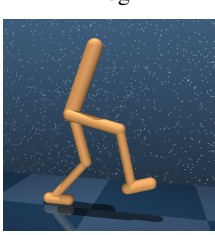

| Fish | Hopper | Humanoid | Walker |

Figure 6: **DeepMind Control Suite benchmarking domains (Tassa et al., 2020).**

Table 5: **Overview of DeepMind Control Suite tasks.** Classification is based on Hubert et al. (2021); Yarats et al. (2022)

| Task | Difficulty | Reward | $\dim(\mathcal{S})$ | $\dim(\mathcal{A})$ |
|---|---|---|---|---|
| Acrobot Swingup | hard | dense | 6 | 1 |
| Cartpole Swingup Sparse | easy | sparse | 5 | 1 |
| Dog Run | hard | dense | 223 | 38 |
| Dog Walk | hard | dense | 223 | 38 |
| Fish Swim | medium | dense | 24 | 5 |
| Hopper Hop | medium | dense | 15 | 4 |
| Humanoid Run | hard | dense | 67 | 24 |
| Humanoid Walk | hard | dense | 67 | 24 |

We consider following eight tasks from Meta-World:

- Assembly: Pick up a nut and place it onto a peg (peg and nut positions are randomized),
- Button Press: Press a button (button positions are randomized),
- Disassemble: Remove a nut from a peg (peg and nut positions are randomized),
- Lever Pull: Pull a lever down 90 degrees (lever positions are randomized),
- Pick Place Wall: Pick a puck, bypass a wall and place the puck (puck and goal positions are randomized),
- Push Back: Push the puck to a goal (puck and goal positions are randomized),
- Shelf Place: Pick and place a puck onto a shelf (puck and shelf positions are randomized),
- Window Open: Push and open a window (window positions are randomized).

All tasks from Meta-World share the same embodiment, observation space ($\dim(\mathcal{S}) = 39$) and action space ($\dim(\mathcal{A}) = 4$). Please refer to Yu et al. (2019) for the definitions of the reward functions and success metrics used in the Meta-World tasks.

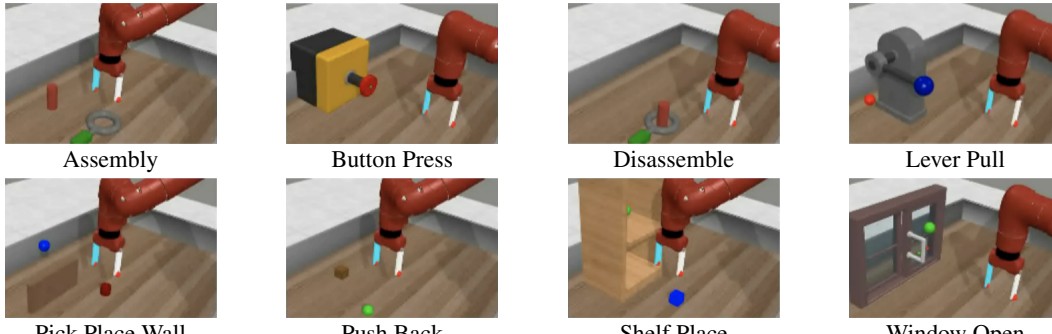

| Assembly | Button Press | Disassemble | Lever Pull |

| Pick Place Wall | Push Back | Shelf Place | Window Open |

Figure 7: **Meta-World manipulation tasks.** We consider eight different tasks from the Meta-World Benchmark.

We further consider following eight tasks from the twelve benchmarking locomotion tasks of HumanoidBench:

- Balance Hard: Balance on the unstable board while the spherical pivot beneath the board does move,
- Balance Simple: Balance on the unstable board while the spherical pivot beneath the board does not move,
- Hurdle: Keep forward velocity close to 5 m/s while successfully overcoming hurdles,
- Maze: Reach the goal position in a maze by taking multiple turns at the intersections,
- Reach: Reach a randomly initialized 3D point with the left hand,
- Run: Run forward at a speed of 5 m/s,
- Slide: Walk over an iterating sequence of upward and downward slides at 1 m/s,
- Stair: Traverse an iterating sequence of upward and downward stairs at 1 m/s.

Visualizations of the tasks are shown in Fig. 8.

The benchmark uses the Unitree H1 with two dexterous hands. The observation and action spaces, and degrees of freedom of the robot system with the dexterous hands are summarized in Tab. 6.

Table 6: **Humanoid robot specifications with two hands.**

| Parameter | Value |
| --- | --- |
| Observation space | 151 |
| Action space | 61 |
| DoF (body) | 25 |
| DoF (hands) | 50 |

## B  IMPLEMENTATION DETAILS

**TD-MPC2 implementation.** We use the official implementation of TD-MPC2 avaliable at `https://github.com/nicklashansen/tdmpc2`, and use the default hyperparameters suggested by the authors. A complete list of hyperparameters is provided in Tab. 7. Details on TD-MPC2 can be found in Appendix B.1.

**BMPC implementation.** We use the official implementation of BMPC from `https://github.com/wertyuilife2/bmpc`, and use the default hyperparameters suggested by the authors.

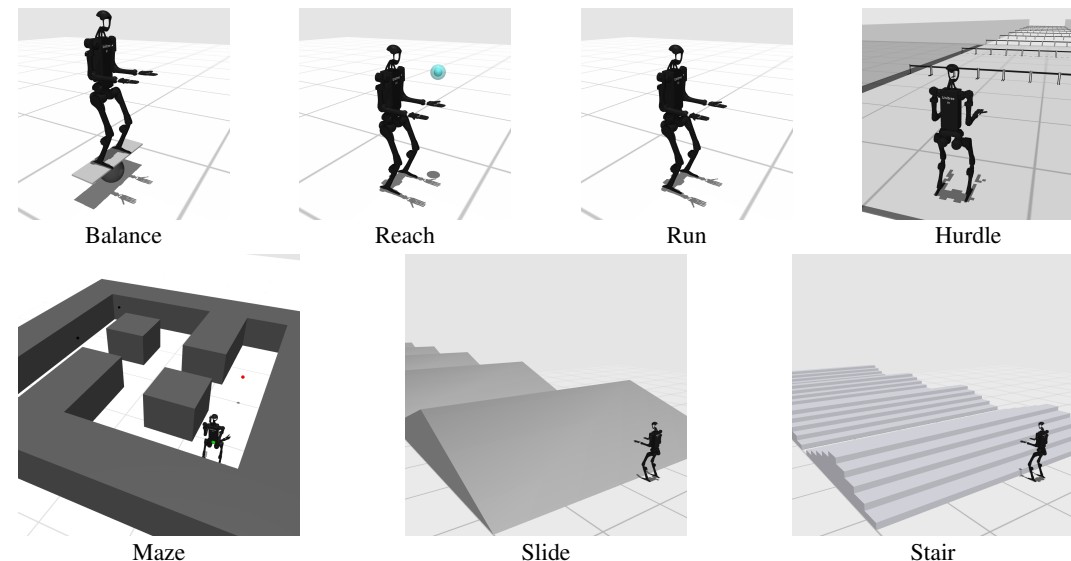

Figure 8: **HumanoidBench locomotion tasks.** We consider eight tasks from the HumanoidBench locomotion benchmark that cover a wide variety of interactions and difficulties. This figure illustrates an initial state for each task.

Since the code is based on the official TD-MPC2 codebase and incorporates both algorithms, we use this implementation as a basis for our method. Details on BMPC are provided in Appendix B.2.

**Dreamer-v3 baseline implementation.** We use the official implementation of Dreamer-v3 available at `https://github.com/danijar/dreamerv3`. We follow the decision of Hansen et al. (2024) and use the authors' suggested hyperparameters for proprioceptive control (DeepMind Control Suite). Please refer to Hafner et al. (2025) and Hansen et al. (2024) for a complete list of hyperparameters and implementation details.

**SAC baseline implementation.** We use the SAC implementation from `https://github.com/denisyarats/pytorch_sac` as in the TD-MPC (Hansen et al., 2022) paper, and use the hyperparameters suggested by the authors. Please refer to their paper for a complete list of hyperparameters.

## B.1 TD-MPC2

**Architectural details.** All components of TD-MPC2 are implemented as multi-layer perceptrons (MLPs). The encoder $h$ contains a variable number of layers $(2-5)$, depending on the architecture size; all other components are 3-layer MLPs. Intermediate layers consist of a linear layer followed by LayerNorm and a Mish activation function. The latent representation is normalized as a simplicial embedding. $Q$-functions additionally use Dropout. We summarize the TD-MPC2 architecture for the 5M parameter `base` (default for online RL) model size using PyTorch-like notation:

```
Encoder parameters: 167,936
Dynamics parameters: 843,264
Reward parameters: 631,397
Policy parameters: 582,668
Q parameters: 3,156,985
Task parameters: 7,680
Total parameters: 5,389,930

Architecture: TD-MPC2 base 5M(
  (task_embedding): Embedding(T, 96, max_norm=1)
  (encoder): ModuleDict(
    (state): Sequential(
      (0): NormedLinear(in_features=S+T, out_features=256, act=Mish)
      (1): NormedLinear(in_features=256, out_features=512, act=SimNorm)
    )
  )
```

```
(dynamics): Sequential(
  (0): NormedLinear(in_features=512+T+A, out_features=512, act=Mish)
  (1): NormedLinear(in_features=512, out_features=512, act=Mish)
  (2): NormedLinear(in_features=512, out_features=512, act=SimNorm)
)
(reward): Sequential(
  (0): NormedLinear(in_features=512+T+A, out_features=512, act=Mish)
  (1): NormedLinear(in_features=512, out_features=512, act=Mish)
  (2): Linear(in_features=512, out_features=101,)
)
(pi): Sequential(
  (0): NormedLinear(in_features=512+T, out_features=512, act=Mish)
  (1): NormedLinear(in_features=512, out_features=512, act=Mish)
  (2): Linear(in_features=512, out_features=2A, bias=True)
)
(Qs): Vectorized ModuleList(
  (0-4): 5 x Sequential(
    (0): NormedLinear(in_features=512+T+A, out_features=512, dropout=0.01, act=Mish)
    (1): NormedLinear(in_features=512, out_features=512, act=Mish)
    (2): Linear(in_features=512, out_features=101, bias=True)
  )
)
```

where `S` is the input dimensionality, `T` is the number of tasks, and `A` is the action space. We exclude the task embedding `T` from single-task experiments. The exact parameter counts listed above are for `S`$= 39$, `T`$= 80$, and `A`$= 6$. Since we only perform single-task experiments in this work, all models contain around 5M parameters for TD-MPC2.

**Policy-guided MPC.** TD-MPC2 uses Model Predictive Path Integral (MPPI) (Williams et al., 2015; 2017) for local trajectory optimization, which is a gradient-free, sampling-based MPC method. MPPI iteratively samples action sequences $(a_t, a_{t+1}, \ldots, a_{t+H})$ of length $H$ from $\mathcal{N}(\mu, \sigma^2)$, evaluates their expected return by rolling out latent trajectories with the model, and updates the parameters $\mu, \sigma$ of a time-dependent multivariate Gaussian with diagonal covariance based on a weighted average such that the expected return is maximized. This iterative optimization procedure is repeated for a fixed number of iterations and the first action $a_t \sim \mathcal{N}(\mu_t^*, \sigma_t^*)$ is applied to the environment. TD-MPC2 augments the sampling procedure with samples from the policy prior $\pi_\theta$ and warm-starts the optimization procedure by initializing $(\mu, \sigma)$ as the solution of the previous step shifted by one to improve performance. Please refer to Hansen et al. (2022) for more details.

## B.2 BMPC

**Architectural details.** The main architectural difference of BMPC to TD-MPC2 is that it uses two $V$-functions instead of five $Q$-functions:

```
V parameters: 1,256,650

Total parameters: 3,489,595

Architecture: Difference BMPC to TD-MPC2
  (
    (Vs): Vectorized ModuleList(
      (0-1): 2 x Sequential(
        (0): NormedLinear(in_features=512+T, out_features=512, dropout=0.01, act=Mish)
        (1): NormedLinear(in_features=512, out_features=512, act=Mish)
        (2): Linear(in_features=512, out_features=101, bias=True)
      )
    )
  )
```

**Model-based TD-learning.** Since BMPC does not use a SAC-style max-Q approach for policy improvement, the authors decide to learn a state value function $V_\phi$ instead of a state-action value function $Q_\phi$. The value network is learned by minimizing the cross-entropy loss with respect to the discretized n-step TD-target $\hat{V}$ computed by using the latest model, policy, and target value network:

$$\mathcal{L}_V(\phi) \doteq \mathop{\mathbb{E}}_{(\mathbf{s},\mathbf{a})_{0:H} \sim \mathcal{B}} \left[ \sum_{t=0}^{H} \lambda^t \left[ \mathrm{CE}(V_\phi(\mathbf{z}_t), \hat{V}(h(\mathbf{s}_t))) \right] \right], \; \mathbf{z}_0 = h(\mathbf{s}_0), \; \mathbf{z}_{t+1} = d(\mathbf{z}_t, \mathbf{a}_t)$$

$$\hat{V}(\mathbf{z}_t') \doteq \gamma^N V_{\phi^-}(\mathbf{z}_{t+N}') + \sum_{k=0}^{N-1} \gamma^k R(\mathbf{z}_{t+k}', \pi_\theta(\mathbf{z}_{t+k}')), \; \mathbf{z}_{t+1}' = d(\mathbf{z}_t', \pi_\theta(\mathbf{z}_t'))$$

(8)

Table 7: **TD-MPC2 hyperparameters.** We use the same hyperparameters across all tasks. Certain hyperparameters are set automatically using heuristics.

| Hyperparameter | Value |
|---|---|
| **Planning** | |
| Horizon ($H$) | 3 |
| Iterations | 6 ($+2$ if $\|\mathcal{A}\| \geq 20$) |
| Population size | 512 |
| Policy prior samples | 24 |
| Number of elites | 64 |
| Minimum std. | 0.05 |
| Maximum std. | 2 |
| Temperature | 0.5 |
| Momentum | No |
| | |
| **Policy prior** | |
| Log std. min. | $-10$ |
| Log std. max. | 2 |
| | |
| **Replay buffer** | |
| Capacity | $1,000,000$ |
| Sampling | Uniform |
| | |
| **Architecture (5M)** | |
| Encoder dim | 256 |
| MLP dim | 512 |
| Latent state dim | 512 |
| Task embedding dim | 96 |
| Task embedding norm | 1 |
| Activation | LayerNorm + Mish |
| $Q$-function dropout rate | 1% |
| Number of $Q$-functions | 5 |
| Number of reward/value bins | 101 |
| SimNorm dim ($V$) | 8 |
| SimNorm temperature ($\tau$) | 1 |
| | |
| **Optimization** | |
| Update-to-data ratio | 1 |
| Batch size | 256 |
| Joint-embedding coef. | 20 |
| Reward prediction coef. | 0.1 |
| Value prediction coef. | 0.1 |
| Temporal coef. ($\lambda$) | 0.5 |
| $Q$-fn. momentum coef. | 0.99 |
| Policy prior entropy coef. | $1 \times 10^{-4}$ |
| Policy prior loss norm. | Moving $(5\%, 95\%)$ percentiles |
| Optimizer | Adam (Kingma & Ba, 2015) |
| Learning rate | $3 \times 10^{-4}$ |
| Encoder learning rate | $1 \times 10^{-4}$ |
| Gradient clip norm | 20 |
| Discount factor | Heuristic |
| Seed steps | Heuristic |

where $N$ is the TD horizon, $\mathbf{z}_{0:H}$ are latent vectors rolled out through the models $h$ and $d$. $\hat{V}$ is the TD-target computed using the model $d, R$ and the policy $\pi_\theta$ in an on-policy manner. The authors use a fixed value of $N = 1$ to keep compounding errors small.

**Lazy reanalyze.** BMPC stores imitation targets in the replay buffer and uses lazy reanalyze to avoid costly replanning for all samples during every update to compute the policy objective. For every $k$-th network update, $b$ samples are drawn from the batch and used to get new imitation targets, i.e., the mean and standard deviation of the action distribution $\pi_t = \pi_{\text{MPC}}(\cdot|h(\mathbf{s}_t), \pi_\theta)$ by replanning. These targets $\pi_t$ are then placed back into the replay buffer. Since the replanning is performed independently of the training process, the replay buffer can be approximately seen as an expert dataset and used to sample state-action pairs from it for supervised learning. During replanning, additional noise is added to the policy prior to increase exploration in MPC planning. Thus, the resulting surrogate policy objective with lazy reanalyze can be defined as:

$$\mathcal{L}_\pi^{\text{lazy}}(\theta) \doteq \mathop{\mathbb{E}}_{(\mathbf{s},\mathbf{a},\pi)_{0:H}\sim B} \left[ \sum_{t=0}^{H} \lambda^t \left[ \text{KL}(\pi_t, \pi_\theta(\cdot|\mathbf{z}_t))/\max(1, S) - \beta \mathcal{H}(\pi_\theta(\cdot|\mathbf{z}_t)) \right] \right] \tag{9}$$

where $\pi_t$ is the expert action distribution from the replay buffer.

Table 8: **BMPC hyperparameters.** We use the same hyperparameters for all tasks. All other hyperparameters are the default TD-MPC2 values.

| Hyperparameter | Value |
|---|---|
| **Planning** | |
| Horizon | 3 |
| Replanning horizon | 3 |
| Lazy reanalyze interval ($k$) | 10 |
| Lazy reanalyze batch size ($b$) | 20 |
| | |
| **Policy prior** | |
| Log std. min. | $-3$ |
| Log std. max. | 1 |
| Log std. min. (replanning) | $-2$ |
| Log std. max. (replanning) | 1 |
| | |
| **Architecture** | |
| Number of $V$-functions | 2 |
| | |
| **Optimization** | |
| Batch size | 256 |
| TD horizon ($N$) | 1 |
| Policy prior entropy coef. | $1 \times 10^{-4}$ |

### B.3 DREAM-MPC

**Hyperparameters.** We use the same hyperparameters across all tasks. The hyperparameters specific to our method are listed in Tab. 1.

## C ADDITIONAL RESULTS

In this section, we provide the learning curves for all baselines as well as detailed evaluation results for all environments.

### C.1 LEARNING CURVES

Figs. 9 to 11 show the episode returns and the success rates as a function of environment steps, respectively.

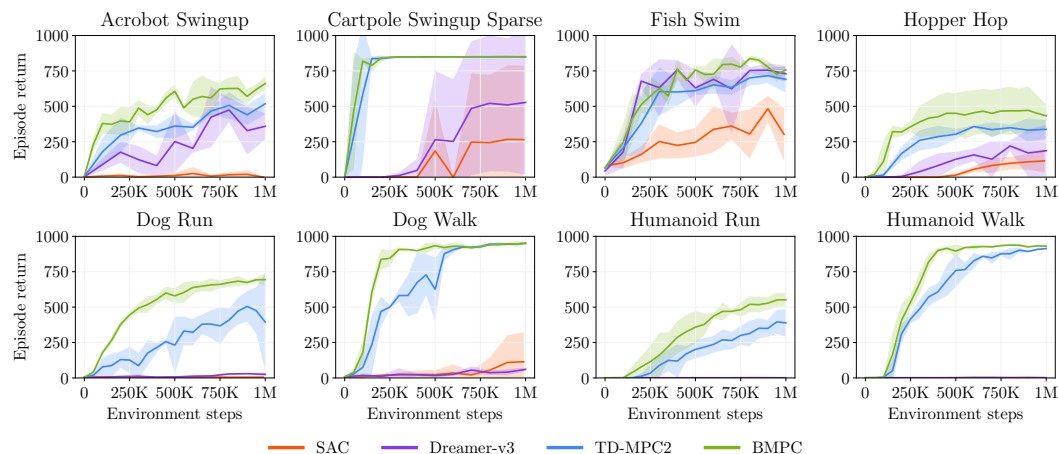

Figure 9: **Learning curves for the DeepMind Control Suite.** The line represents the mean episodic return and the shaded area the 95% confidence interval across 3 seeds.

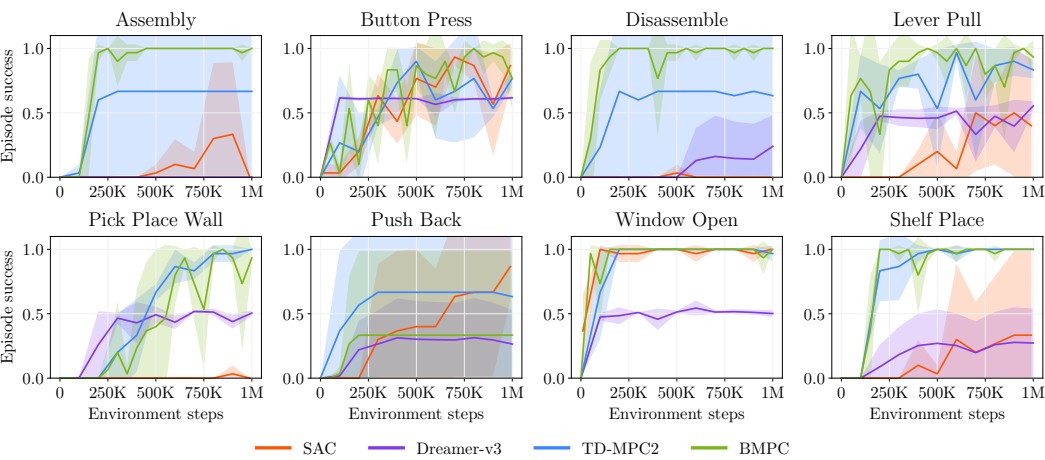

Figure 10: **Learning curves for Meta-World.** The line represents the mean episodic return and the shaded area the 95% confidence interval across 3 seeds.

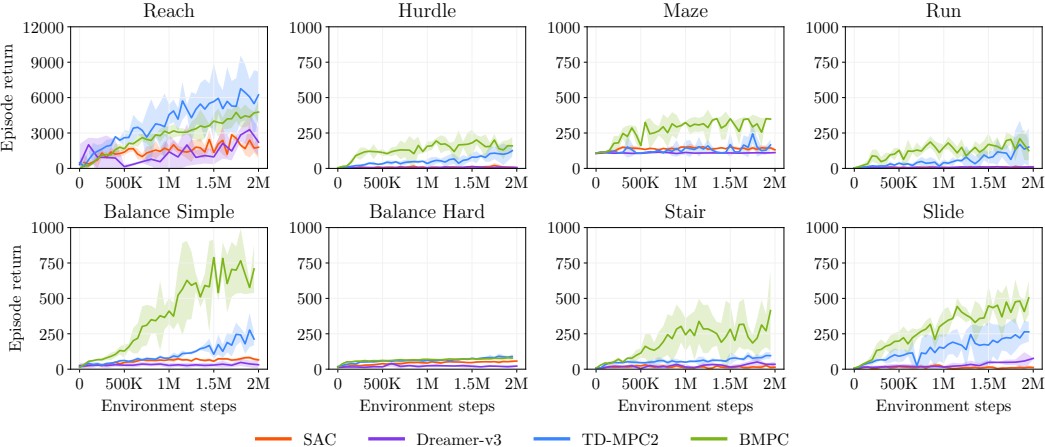

Figure 11: **Learning curves for HumanoidBench.** The line represents the mean episodic return and the shaded area the 95% confidence interval across 3 seeds.

## C.2 DETAILED EVALUATION RESULTS

We find that having a good policy is important because it leads to better value estimates, which are crucial for gradient-based MPC. While Dream-MPC can improve the performance of the policy for TD-MPC2, it cannot consistently match the performance of MPPI. Since the performance of the policy is quite weak as shown in Tabs. 12 to 14, this fact favours MPPI, which has a higher diversity of the initial solutions due to the sampling procedure. While we can further improve the performance of Dream-MPC with TD-MPC2 as a basis, for example by increasing the number of optimization iterations, this also increases computational costs. This highlights the importance of a good initial solution to warm-start the MPC optimization process, especially for high-dimensional problems.

Table 9: **DeepMind Control Suite evaluation results of different algorithms.**

| Task | SAC | Dreamer-v3 | TD-MPC2 | BMPC | Dream-MPC (TD-MPC2) | Dream-MPC (BMPC) |
|---|---|---|---|---|---|---|
| Acrobot Swingup | $176 \pm 21$ | $372 \pm 141$ | $\underline{595 \pm 34}$ | $587 \pm 25$ | $590 \pm 40$ | $\mathbf{596 \pm 50}$ |
| Cartpole Swingup Sparse | $788 \pm 10$ | $538 \pm 325$ | $\underline{848 \pm 0}$ | $837 \pm 14$ | $847 \pm 3$ | $\mathbf{849 \pm 1}$ |
| Fish Swim | $657 \pm 110$ | $729 \pm 98$ | $786 \pm 8$ | $\underline{804 \pm 17}$ | $764 \pm 56$ | $\mathbf{816 \pm 11}$ |
| Hopper Hop | $287 \pm 15$ | $198 \pm 111$ | $\mathbf{493 \pm 47}$ | $404 \pm 39$ | $307 \pm 38$ | $\underline{423 \pm 54}$ |
| Dog Run | $15 \pm 6$ | $26 \pm 7$ | $358 \pm 228$ | $\underline{678 \pm 27}$ | $115 \pm 72$ | $\mathbf{703 \pm 19}$ |
| Dog Walk | $42 \pm 33$ | $47 \pm 20$ | $933 \pm 10$ | $\underline{937 \pm 4}$ | $389 \pm 22$ | $\mathbf{946 \pm 7}$ |
| Humanoid Run | $83 \pm 43$ | $1 \pm 1$ | $344 \pm 60$ | $\underline{528 \pm 29}$ | $110 \pm 10$ | $\mathbf{531 \pm 38}$ |
| Humanoid Walk | $364 \pm 95$ | $2 \pm 1$ | $899 \pm 10$ | $\underline{917 \pm 6}$ | $338 \pm 63$ | $\mathbf{937 \pm 4}$ |
| **Mean** | $302 \pm 269$ | $239 \pm 261$ | $657 \pm 225$ | $\underline{711 \pm 181}$ | $433 \pm 259$ | $\mathbf{725 \pm 181}$ |

The results are the mean episode returns and standard deviations for three random seeds and ten test episodes. **Best** and second best results are highlighted.

Table 10: **Meta-World evaluation results of different algorithms.**

| Task | SAC | Dreamer-v3 | TD-MPC2 | BMPC | Dream-MPC (TD-MPC2) | Dream-MPC (BMPC) |
|---|---|---|---|---|---|---|
| Assembly | $0.0 \pm 0.0$ | $0.0 \pm 0.0$ | $1.0 \pm 0.0$ | $1.0 \pm 0.0$ | $\underline{1.0 \pm 0.0}$ | $\mathbf{1.0 \pm 0.0}$ |
| Button Press | $0.27 \pm 0.31$ | $\underline{0.61 \pm 0.02}$ | $0.33 \pm 0.47$ | $0.33 \pm 0.47$ | $0.33 \pm 0.47$ | $\mathbf{0.67 \pm 0.47}$ |
| Disassemble | $0.03 \pm 0.05$ | $0.27 \pm 0.23$ | $0.67 \pm 0.47$ | $\underline{1.0 \pm 0.0}$ | $0.67 \pm 0.47$ | $\mathbf{1.0 \pm 0.0}$ |
| Lever Pull | $0.03 \pm 0.05$ | $0.52 \pm 0.1$ | $0.0 \pm 0.0$ | $\underline{0.67 \pm 0.47}$ | $0.0 \pm 0.0$ | $\mathbf{0.67 \pm 0.47}$ |
| Pick Place Wall | $0.0 \pm 0.0$ | $0.21 \pm 0.24$ | $\mathbf{1.0 \pm 0.0}$ | $0.0 \pm 0.0$ | $0.67 \pm 0.47$ | $\underline{0.67 \pm 0.47}$ |
| Push Back | $0.67 \pm 0.47$ | $0.32 \pm 0.23$ | $\underline{0.67 \pm 0.47}$ | $0.33 \pm 0.47$ | $\mathbf{0.67 \pm 0.47}$ | $0.33 \pm 0.47$ |
| Shelf Place | $0.0 \pm 0.0$ | $0.27 \pm 0.21$ | $0.67 \pm 0.47$ | $0.67 \pm 0.47$ | $\underline{1.0 \pm 0.0}$ | $\mathbf{1.0 \pm 0.0}$ |
| Window Open | $1.0 \pm 0.0$ | $0.48 \pm 0.09$ | $\underline{1.0 \pm 0.0}$ | $0.67 \pm 0.47$ | $0.67 \pm 0.47$ | $\mathbf{1.0 \pm 0.0}$ |
| **Mean** | $0.25 \pm 0.36$ | $0.33 \pm 0.18$ | $\underline{0.67 \pm 0.33}$ | $0.58 \pm 0.32$ | $0.62 \pm 0.31$ | $\mathbf{0.79 \pm 0.23}$ |

The results are the mean episode successes and standard deviations for three random seeds and ten test episodes. **Best** and second best results are highlighted.

Table 11: **HumanoidBench evaluation results of different algorithms.**

| Task | SAC | Dreamer-v3 | TD-MPC2 | BMPC | Dream-MPC (TD-MPC2) | Dream-MPC (BMPC) |
|---|---|---|---|---|---|---|
| Balance Hard | $55 \pm 3$ | $28 \pm 12$ | $\mathbf{92 \pm 12}$ | $81 \pm 12$ | $45 \pm 10$ | $\underline{82 \pm 12}$ |
| Balance Simple | $70 \pm 10$ | $39 \pm 14$ | $240 \pm 37$ | $\underline{489 \pm 84}$ | $47 \pm 14$ | $\mathbf{654 \pm 89}$ |
| Hurdle | $5 \pm 3$ | $13 \pm 5$ | $78 \pm 24$ | $\underline{120 \pm 43}$ | $12 \pm 1$ | $\mathbf{249 \pm 34}$ |
| Maze | $140 \pm 7$ | $110 \pm 4$ | $169 \pm 47$ | $\mathbf{349 \pm 2}$ | $120 \pm 8$ | $\underline{266 \pm 33}$ |
| Reach | $2048 \pm 212$ | $2151 \pm 1038$ | $\mathbf{5037 \pm 1436}$ | $4125 \pm 324$ | $2751 \pm 444$ | $\underline{4348 \pm 215}$ |
| Run | $8 \pm 3$ | $11 \pm 5$ | $136 \pm 110$ | $\underline{139 \pm 81}$ | $10 \pm 7$ | $\mathbf{302 \pm 11}$ |
| Slide | $11 \pm 5$ | $56 \pm 29$ | $237 \pm 54$ | $\underline{442 \pm 36}$ | $16 \pm 3$ | $\mathbf{632 \pm 114}$ |
| Stair | $15 \pm 15$ | $35 \pm 17$ | $100 \pm 18$ | $\underline{403 \pm 145}$ | $30 \pm 6$ | $\mathbf{456 \pm 145}$ |
| **Mean** | $294 \pm 664$ | $305 \pm 698$ | $761 \pm 1617$ | $\underline{769 \pm 1277}$ | $379 \pm 897$ | $\mathbf{874 \pm 1326}$ |

The results are the mean episode returns and standard deviations for three random seeds and ten test episodes. **Best** and second best results are highlighted.

## C.3 DETAILED TD-MPC2 AND BMPC RESULTS

We include full results of TD-MPC2 and BMPC for all environments in Tabs. 12 to 14, including the performance of using the underlying policy network only. We also conduct experiments in which we apply the test-time regularization defined in Eq. (5) with a regularization coefficient of $\lambda_{\mathrm{unc}} = 0.01$ to TD-MPC2 and BMPC. While the regularization can improve the performance of BMPC in some cases, it causes a significant performance decrease for TD-MPC2, especially for high-dimensional problems.

Table 12: **DeepMind Control Suite evaluation results of different TD-MPC2 and BMPC variants.**

| Environment | TD-MPC2 | TD-MPC2 (policy only) | TD-MPC2 (w/ test-time regularization) | BMPC | BMPC (policy only) | BMPC (w/ test-time regularization) |
|---|---|---|---|---|---|---|
| Acrobot Swingup | **595 ± 34** | 551 ± 21 | 594 ± 32 | 587 ± 25 | 564 ± 52 | 573 ± 11 |
| Cartpole Swingup Sparse | 848 ± 0 | 760 ± 114 | 848 ± 0 | 837 ± 14 | **848 ± 1** | 845 ± 3 |
| Fish Swim | 786 ± 8 | 645 ± 83 | 783 ± 13 | 804 ± 17 | **804 ± 14** | 776 ± 9 |
| Hopper Hop | **493 ± 47** | 383 ± 154 | 465 ± 79 | 404 ± 39 | 445 ± 106 | 440 ± 87 |
| Dog Run | 358 ± 228 | 89 ± 52 | 376 ± 231 | 678 ± 27 | 670 ± 13 | **678 ± 23** |
| Dog Walk | 933 ± 10 | 298 ± 20 | 926 ± 9 | 937 ± 4 | 930 ± 5 | **940 ± 4** |
| Humanoid Run | 344 ± 60 | 65 ± 2 | 345 ± 55 | **528 ± 29** | 458 ± 15 | 514 ± 31 |
| Humanoid Walk | 899 ± 10 | 142 ± 36 | 881 ± 9 | 917 ± 6 | 930 ± 7 | **931 ± 3** |
| **Mean** | 657 ± 225 | 367 ± 247 | 652 ± 221 | 711 ± 181 | 706 ± 187 | **712 ± 179** |

The results are the mean episode returns and standard deviations for three random seeds and ten test episodes. **Best** and second best results are highlighted.

Table 13: **Meta-World evaluation results of different TD-MPC2 and BMPC variants.**

| Environment | TD-MPC2 | TD-MPC2 (policy only) | TD-MPC2 (w/ test-time regularization) | BMPC | BMPC (policy only) | BMPC (w/ test-time regularization) |
|---|---|---|---|---|---|---|
| Assembly | 1.0 ± 0.0 | 1.0 ± 0.0 | 0.67 ± 0.47 | 1.0 ± 0.0 | 1.0 ± 0.0 | **1.0 ± 0.0** |
| Button Press | 0.33 ± 0.47 | 0.0 ± 0.0 | 0.67 ± 0.47 | 0.33 ± 0.47 | **1.0 ± 0.0** | 0.33 ± 0.47 |
| Disassemble | 0.67 ± 0.47 | 0.67 ± 0.47 | 0.67 ± 0.47 | 1.0 ± 0.0 | 0.67 ± 0.47 | **1.0 ± 0.0** |
| Lever Pull | 0.0 ± 0.0 | 0.0 ± 0.0 | 0.0 ± 0.0 | 0.67 ± 0.47 | **1.0 ± 0.0** | 0.67 ± 0.47 |
| Pick Place Wall | **1.0 ± 0.0** | 0.0 ± 0.0 | 0.33 ± 0.47 | 0.0 ± 0.0 | 0.67 ± 0.47 | 0.33 ± 0.47 |
| Push Back | 0.67 ± 0.47 | 0.33 ± 0.47 | **0.67 ± 0.47** | 0.33 ± 0.47 | 0.33 ± 0.47 | 0.33 ± 0.47 |
| Shelf Place | 0.67 ± 0.47 | 0.67 ± 0.47 | 1.0 ± 0.0 | 0.67 ± 0.47 | 1.0 ± 0.0 | **1.0 ± 0.0** |
| Window Open | 1.0 ± 0.0 | 0.33 ± 0.47 | 1.0 ± 0.0 | 0.67 ± 0.47 | **1.0 ± 0.0** | 0.67 ± 0.47 |
| **Mean** | 0.67 ± 0.33 | 0.38 ± 0.35 | 0.62 ± 0.31 | 0.58 ± 0.32 | **0.83 ± 0.24** | 0.67 ± 0.29 |

The results are the mean episode returns and standard deviations for three random seeds and ten test episodes. **Best** and second best results are highlighted.

Table 14: **HumanoidBench evaluation results of different TD-MPC2 and BMPC variants.**

| Environment | TD-MPC2 | TD-MPC2 (policy only) | TD-MPC2 (w/ test-time regularization) | BMPC | BMPC (policy only) | BMPC (w/ test-time regularization) |
|---|---|---|---|---|---|---|
| Balance Hard | 92 ± 12 | 34 ± 3 | **94 ± 22** | 81 ± 12 | 78 ± 8 | 80 ± 9 |
| Balance Simple | 240 ± 37 | 33 ± 16 | 208 ± 34 | 489 ± 84 | 414 ± 45 | **778 ± 77** |
| Hurdle | 78 ± 24 | 14 ± 3 | 73 ± 27 | 120 ± 43 | 147 ± 40 | **175 ± 51** |
| Maze | 169 ± 47 | 111 ± 3 | 115 ± 4 | **349 ± 2** | 121 ± 7 | 347 ± 4 |
| Reach | **5037 ± 1436** | 1558 ± 368 | 399 ± 208 | 4125 ± 324 | 2117 ± 309 | 2279 ± 376 |
| Run | 136 ± 110 | 8 ± 4 | 99 ± 72 | 139 ± 81 | 91 ± 25 | **222 ± 56** |
| Slide | 237 ± 54 | 14 ± 2 | 248 ± 77 | 442 ± 36 | 250 ± 26 | **553 ± 100** |
| Stair | 100 ± 18 | 24 ± 8 | 91 ± 23 | 403 ± 145 | 208 ± 46 | **432 ± 199** |
| **Mean** | 761 ± 1617 | 224 ± 505 | 166 ± 106 | **769 ± 1277** | 428 ± 646 | 608 ± 665 |

The results are the mean episode returns and standard deviations for three random seeds and ten test episodes. **Best** and second best results are highlighted.

## D  INTEGRATION INTO DREAMER

We further integrate our base method (without uncertainty regularization) into Dreamer (Hafner et al., 2020) to show that it also works with other model-based RL algorithms. Dreamer learns a latent dynamics model, often referred to as a world model, consisting of the following components:

- Representation model: $p_\theta(s_t|s_{t-1}, a_{t-1}, o_t)$

- Transition model: $q_\theta(s_t|s_{t-1}, a_{t-1})$

- Reward model: $q_\theta(r_t|s_t)$

- Observation model (only used as an additional learning signal): $q_\theta(o_t|s_t)$

All components are jointly optimized to increase the variational lower bound (ELBO), including reconstruction terms for observations and rewards as well as a KL regularizer:

$$\mathcal{L}_{\text{Rec}} = \mathbb{E}\left[\sum_t (\mathcal{L}_O^t + \mathcal{L}_R^t + \mathcal{L}_D^t)\right] + \text{const}, \tag{10}$$

where

$$\begin{aligned}
\mathcal{L}_O^t &= \ln q(o_t|s_t), \\
\mathcal{L}_R^t &= \ln q(r_t|s_t), \\
\mathcal{L}_D^t &= -\beta \text{KL}(p(s_t|s_{t-1}, a_{t-1}, o_t)||q(s_t|s_{t-1}, a_{t-1})).
\end{aligned} \tag{11}$$

The expected values are calculated based on the dataset and representation model. Please refer to Hafner et al. (2020) for the derivation of the variational bound.

Following the original Dreamer implementation, we estimate state values using $V_\lambda$, an exponentially-weighted average of the reward estimates for a different number of steps beyond the horizon with the learned value model to balance bias and variance:

$$V_R(s_\tau) = \mathbb{E}_{q_\theta, \pi_\phi}\left[\sum_{n=\tau}^{t+H} r_n\right], \tag{12}$$

$$V_N^k(s_\tau) = \mathbb{E}_{q_\theta, \pi_\phi}\left[\sum_{n=\tau}^{h-1} \gamma^{n-\tau} r_n + \gamma^{h-\tau} v_\psi(s_h)\right] \quad \text{with } h = \min(\tau + k, t + H), \tag{13}$$

$$V_\lambda(s_\tau) = (1-\lambda)\sum_{n=1}^{H-1} \lambda^{n-1} V_N^n(s_\tau) + \lambda^{H-1} V_N^H(s_\tau). \tag{14}$$

For each time step $t$, Dream-MPC creates an initial sequence of actions by performing an imaginary rollout of the policy $\pi_\phi$ and generates $N$ candidate trajectories adding small perturbations to the initial action sequence:

$$\{\hat{a}^{(n)}\}_{n=1}^N = \{\pi_\phi(a_{\tau-1}|s_{\tau-1}) + \epsilon_\tau^{(n)}|\tau = t+1, ..., t+H+1\}_{n=1}^N, \quad \text{where } \epsilon_\tau^{(n)} \sim \mathcal{N}(0, \sigma_a^2). \tag{15}$$

The imaginary rollout is done by encoding observations and actions into latent space using the representation model $p_\theta$ and repeatedly calling the one-step transition model $q_\theta$ to generate a sequence of predicted states $\{s_\tau\}_{\tau=t+1}^{t+H+1}$ for each candidate trajectory.

$$s_t^{(n)} \sim p_\theta(s_t^{(n)}|s_{t-1}^{(n)}, a_{t-1}^{(n)}, o_t), \qquad s_{t+1:t+H+1}^{(n)} \sim \prod_{\tau=t+1}^{t+H+1} q_\theta(s_\tau^{(n)}|s_{\tau-1}^{(n)}, a_{\tau-1}^{(n)}) \tag{16}$$

We integrate our gradient-based MPC method into Dreamer as shown in Alg. 2.

---

**Algorithm 2: Dream-MPC integration into Dreamer**

---

**Input:** Representation model $p_\theta(s_t|s_{t-1}, a_{t-1}, o_t)$, transition model $q_\theta(s_t|s_{t-1}, a_{t-1})$, reward model
$q_\theta(r_t|s_t)$, value function model $v_\psi(s_t)$, policy model $\pi_\phi(a_t|s_t)$, exploration noise $p(\epsilon)$, action
repeat $R$, seed episodes $S$, collect interval $C$, batch size $B$, chunk length $L$, learning rate $\eta$

Initialize dataset $\mathcal{D}$ with $S$ random seed episodes.
Initialize model parameters $\theta, \phi, \psi$ randomly.
**while** *not converged* **do**
 **for** *update step* $s = 1..C$ **do**
  // Dynamics model learning
  Draw sequences $\{(o_t, a_t, r_t)_{t=k}^{L+k}\}_{i=1}^B \sim \mathcal{D}$ uniformly at random from the dataset.
  Compute loss $\mathcal{L}(\theta)$ from Eq. (10).
  Update model parameters $\theta \leftarrow \theta - \eta\nabla_\theta\mathcal{L}(\theta)$.
  // Policy learning
  Imagine trajectories $\{(s_\tau, a_\tau)\}_{\tau=t}^{t+H}$ from each $s_t$.
  Predict rewards $\mathbb{E}[q_\theta(r_\tau|s_\tau)]$ and values $v_\psi(s_\tau)$.
  Compute value estimates $V_\lambda(s_\tau)$ via Eq. (14).
  Update $\phi \leftarrow \phi + \eta\nabla_\phi \sum_{\tau=t}^{t+H} V_\lambda(s_\tau)$.
  Update $\psi \leftarrow \psi - \eta\nabla_\psi \sum_{\tau=t}^{t+H} \frac{1}{2}||v_\psi(s_\tau) - V_\lambda(s_\tau)||^2$.

 // Data collection
 $o_1 \leftarrow$ env.reset()
 **for** *time step* $t = 1..[\frac{T}{R}]$ **do**
  Infer current state $s_t \sim p_\theta(s_t|s_{t-1}, a_{t-1}, o_t)$ from the history.
  $a_t \leftarrow$ planner($s_t$), see Alg. 3 for details.
  Add exploration noise $\epsilon \sim p(\epsilon)$ to the action.
  **for** *action repeat* $k = 1..R$ **do**
   $r_t^k, o_{t+1}^k \leftarrow$ env.step($a_t$)
  $r_t, o_{t+1} \leftarrow \sum_{k=1}^R r_t^k, o_{t+1}^R$
 $\mathcal{D} \leftarrow \mathcal{D} \cup \{(o_t, a_t, r_t)_{t=1}^T\}$

---

**Algorithm 3: Dream-MPC planner for Dreamer**

---

**Input:** Representation model $p_\theta(s_t|s_{t-1}, a_{t-1}, o_t)$, transition model $q_\theta(s_t|s_{t-1}, a_{t-1})$, reward model
$q_\theta(r_t|s_t)$, value function model $v_\psi(s_t)$, policy model $\pi_\phi(a_t|s_t)$, planning horizon $H$,
optimization iterations $I$, candidates per iteration $J$, action noise $\sigma_a^2$, action optimization rate $\alpha$

Initialize proposal by rolling out the policy $\pi_\phi$ with the transition model $\hat{a}_{t:t+H} \sim \pi_\phi(s_{t:t+H})$.
Generate $N$ candidates by adding noise $\mathcal{N}(0, \sigma_a^2)$ to the proposal via Eq. (15).
Initialize candidate action sequences $a_{t:t+H}$ via Eq. (3).
**for** *optimization iteration* $i = 1, 2, \dots I$ **do**
 **for** *candidate action sequence* $n = 1, 2, \dots N$ **do**
  Predict imagined states $s_\tau^{(n)} = s_{t:t+H+1}^{(n)}$ via Eq. (16)
  Predict rewards $\mathbb{E}\left[q_\theta(r_\tau^{(n)}|s_\tau^{(n)})\right]$ and values $v_\psi(s_\tau^{(n)})$
  Compute value estimates $V_\lambda(s_\tau^{(n)})$ via Eq. (14)
  Optimize action sequence via $a_\tau^{(n)} \leftarrow \{a_\tau^{(n)} + \alpha\nabla_{a_\tau^{(n)}} V_\lambda^{(n)}(s_\tau^{(n)})|\tau = t, ..., t+H\}$

**Output:** First optimized action $a_t^{(k)}$ with $k = \arg\max_n\{V_\lambda^{(n)}\}_{n=1}^N$.

---

## D.1 EXPERIMENTS

We evaluate our method on four different environments from the DeepMind Control Suite and com-
pare our method with PlaNet (Hafner et al., 2019), Dreamer (Hafner et al., 2020), SAC+AE (Yarats
et al., 2021), a variant of the model-free Soft Actor Critic (SAC) (Haarnoja et al., 2018) algorithm
for image-based observations and the (hybrid) Grad-MPC method proposed in (S V et al., 2023).
Note that hybrid Grad-MPC and Dream-MPC both share the general idea of using a policy network

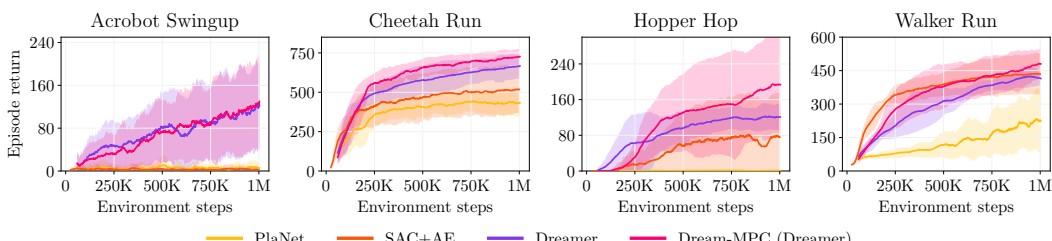

Figure 12: **Learning curves for four tasks from the DeepMind Control Suite.** The line represents the mean episodic return and the shaded area the 95% confidence interval across 3 seeds.

to warm-start gradient-based MPC. We provide a summary of the main differences in Appendix E. All experiments are performed with only RGB visual observations with a resolution of $64 \times 64$.

We evaluate the performance of our method when enabling planning already during training. The learning curves are shown in Fig. 12 and the evaluation results are presented in Tab. 15. We find that our method can not only outperform the baselines, but also that planning during training can improve the sample efficiency without leading to premature convergence. In contrast to PlaNet (CEM) and Grad-MPC, which both use $1000 \times 10 \times 12 = 120\,000$ evaluations of the world model at each time step, our method only requires $5 \times 1 \times 15 = 75$ evaluations. These results are not only promising since Dreamer uses a recurrent dynamics model and a relatively long planning horizon, but also in particular for Acrobot Swingup, which is a non-linear system with chaotic dynamics. All aspects usually affect gradient quality negatively, especially since first order gradient estimators can accumulate significant variance over long-horizon rollouts, which makes them in particular ineffective in chaotic systems (Suh et al., 2022).

Table 15: **Performance comparison of different algorithms.**

| Method | Acrobot Swingup | Cheetah Run | Hopper Hop | Walker Run |
|---|---|---|---|---|
| SAC+AE | $7 \pm 19$ | $495 \pm 100$ | $86 \pm 75$ | $453 \pm 69$ |
| PlaNet | $7 \pm 18$ | $535 \pm 70$ | $1 \pm 4$ | $228 \pm 149$ |
| Dreamer | $134 \pm 91$ | $\underline{751 \pm 111}$ | $\underline{182 \pm 43}$ | $575 \pm 33$ |
| Grad-MPC | $7 \pm 18$ | $438 \pm 81$ | $3 \pm 5$ | $382 \pm 35$ |
| Hybrid Grad-MPC | $\underline{144 \pm 7}$ | $591 \pm 131$ | $158 \pm 47$ | $556 \pm 33$ |
| CEM + policy | $12 \pm 26$ | $674 \pm 20$ | $43 \pm 42$ | $\mathbf{638 \pm 21}$ |
| Dream-MPC (Dreamer) | $\mathbf{147 \pm 101}$ | $\mathbf{836 \pm 49}$ | $\mathbf{298 \pm 86}$ | $\underline{632 \pm 52}$ |

The results are the mean episode returns and standard deviations for three random seeds and ten test episodes. **Best** and underline{second best} results are highlighted.

We benchmark inference times of the different methods on a single Nvidia GeForce RTX 4090 GPU. The results in Tab. 16 show that Dream-MPC is significantly faster as Grad-MPC, which uses a much higher number of candidate trajectories. While hybrid Grad-MPC is faster than Dream-MPC due to using a horizon of one, the overall performance is worse compared to using the policy only because such a myopic optimization is most likely unsuitable for many problems. Note that at the moment a batched version of one operation in the recurrent world model is missing in PyTorch, which slows the parallelized gradient computation down. While this can potentially be further improved, it affects all gradient-based MPC methods in the same way, thus leading to a fair comparison.

Table 16: **Inference times of different methods for Acrobot Swingup.** Mean and standard deviation for three random seeds and ten test episodes per seed.

| Method | Inference time |
|---|---|
| PlaNet | $31.10 \pm 0.65$ ms |
| Grad-MPC | $195.75 \pm 1.33$ ms |
| Hybrid Grad-MPC | $23.16 \pm 0.55$ ms |
| Dream-MPC (Dreamer) | $44.86 \pm 0.60$ ms |

## D.2 GRADIENT ANALYSIS

We evaluate the planner gradients of Grad-MPC and of our method for the ground truth dynamics (simulator) and the learned dynamics model for different planning horizons on the Pendulum-v1 environment with state observations. As Fig. 13 shows, the magnitudes of the gradients are in reasonable orders when using the ground truth dynamics. While the variance increases for longer horizons and might also do for more complex problems, the gradients do not explode or vanish in this case. However, the variance increases significantly for longer planning horizons when using the learned dynamics model. In contrast to Grad-MPC, the variance increases much less for Dream-MPC and although relatively large remains bounded, suggesting that the performance issues of gradient-based planning should not solely be attributed to issues with the gradients caused by the architecture of the world model. Our work shows that there are more aspects that need to be considered such as the quality of the initial proposal for MPC and the learned world model, advocating that further research on gradient-based planning is needed.

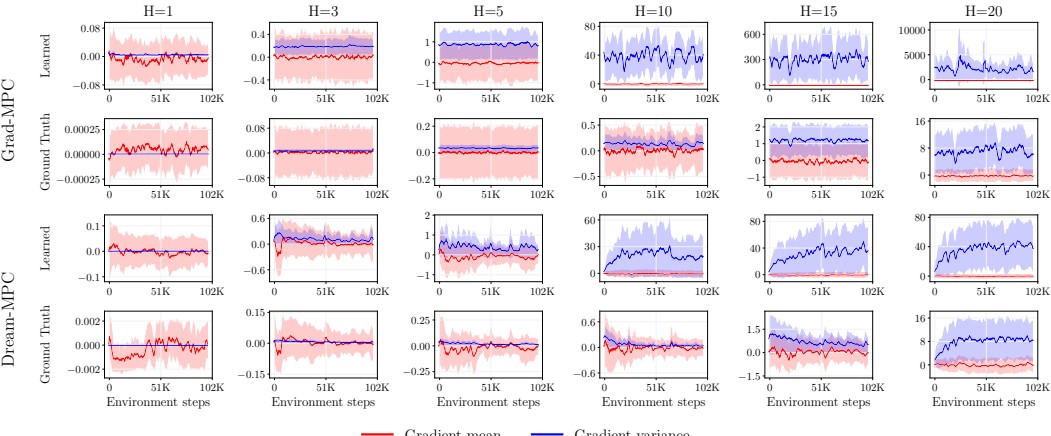

Figure 13: **Planner gradients of Grad-MPC and Dream-MPC.** For different planning horizons on the Pendulum-v1 environment using the ground truth (simulator) and learned dynamics model respectively and state observations. The values are represented by their mean and standard deviation for three different random seeds. The default hyperparameters provided in Tab. 17 are used unless otherwise specified.

As pointed out in Parmas et al. (2023), simply evaluating the gradient quality based on variance alone is insufficient. Thus, we follow the proposal of the authors and analyze the gradients using their Expected Signal-to-Noise Ratio (ESNR), which is defined as

$$\mathrm{ESNR}(\nabla R) = \mathbb{E}\left[\frac{\sum \mathbb{E}[\nabla R]^2}{\sum \mathrm{Var}[\nabla R]}\right], \tag{17}$$

where $R = \sum_{\tau=t+1}^{t+H+1} r_\tau$ is the return, i.e., the undiscounted sum of rewards.

Fig. 14 shows the ESNRs of Grad-MPC and Dream-MPC using the ground truth dynamics or learned dynamics model. While the ESNR remains stable when using the ground truth dynamics, especially for longer horizons the ESNR drops when using the learned model. Recent findings (Georgiev et al., 2025) suggest that learned models can improve ESNR compared to using the ground truth dynamics for some problems, indicating the possibility of further improvement. While the ESNR significantly suffers for horizons greater than ten for Grad-MPC using the learned dynamics model, the ESNR for Dream-MPC remains much more stable for increasing horizons. Together with the variance which increases but does not explode, this suggests that our method is more robust compared to Grad-MPC.

## D.3 MODEL EXPLOITATION

We further analyze the problem of model exploitation, a general challenge in model-based reinforcement learning, where policies tend to exploit inaccuracies in high-capacity dynamics models,

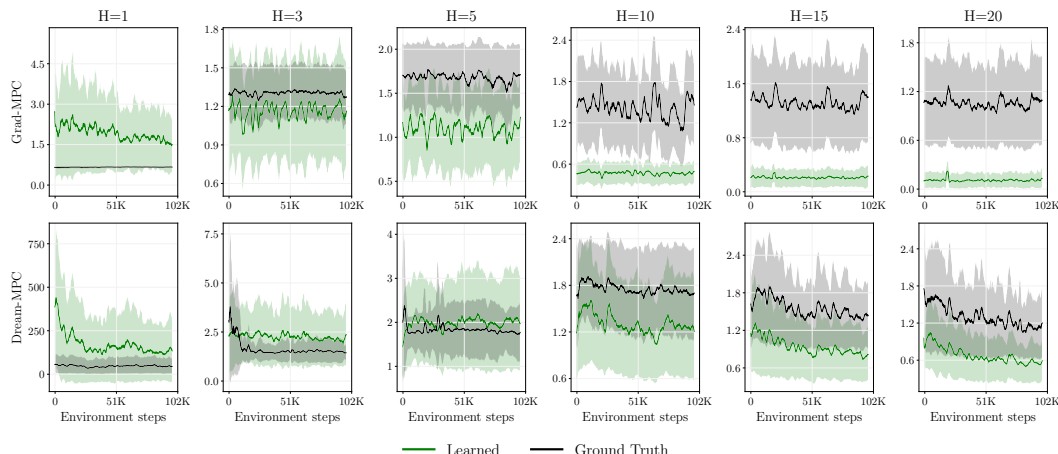

Figure 14: **Expected Signal-to-Noise Ratio (ESNR) of the planner gradients of Grad-MPC and Dream-MPC.** Calculated via Eq. (17) for different planning horizons on the Pendulum-v1 environment using the ground truth (simulator) and learned dynamics model respectively and state observations. The values are represented by their mean and standard deviation for three different random seeds. The default hyperparameters provided in Tab. 17 are used unless otherwise specified.

potentially leading to poor real-world performance despite high predicted returns (Clavera et al., 2018). Since our method optimizes actions to maximize expected returns, we rely on accurate predictions. Fig. 15 shows the mean difference between the actual returns and the predicted returns of a trained policy on the Acrobot Swingup task in for three different seeds and ten test episodes per seed. We find that the differences are quite small, which indicates that the policy may not exploit the learned model. This is probably because the prediction horizon is sufficiently short and MPC may also help to compensate for model inaccuracies by replanning at each step. While the models for other environments might not necessarily be as accurate as for Acrobot Swingup, we empirically find that the learned model tends to estimate the reward quite accurately. Using an ensemble of models to consider uncertainty as for TD-MPC2 can further help to reduce model exploitation.

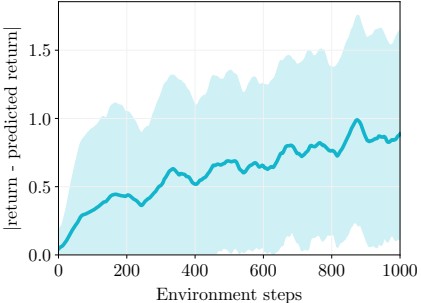

(a) Mean difference between actual and predicted returns and standard deviation for three different seeds and ten test episodes per seed.

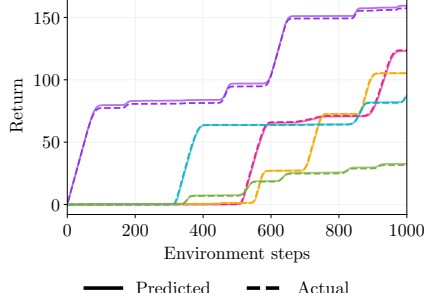

(b) Actual and predicted return for five exemplary evaluation episodes.

Figure 15: **Analysis of predicted returns over the number of environment steps for Acrobot Swingup.**

## D.4 IMPLEMENTATION DETAILS

We use PyTorch (Paszke et al., 2019) implementations of SAC+AE[3], PlaNet and Dreamer[4] that are distributed under MIT license and also base the implementations of hybrid Grad-MPC and of our method on the latter. The hyperparameters are listed in Tab. 17.

We use the default hyperparameters for SAC+AE as described in Yarats et al. (2021), except for the action repeat, which we set to two for a fair comparison.

Table 17: **Hyperparameters and their values used for the experiments.**

| Algorithm | Hyperparameter | Value |
|---|---|---|
| | Optimizer | Adam (Kingma & Ba, 2015) |
| | Max. episode length | 1000 |
| | Action repeat | 2 |
| | Experience size | 1000000 |
| | Embedding size | 1024 |
| | Hidden size | 200 |
| | Belief size | 200 |
| | State size | 30 |
| All | Exploration noise | 0.3 |
| | Seed episodes | 5 |
| | Collect interval | 100 |
| | Batch size | 50 |
| | Overshooting distance | 0 |
| | Overshooting KL beta | 0 |
| | Overshooting reward scale | 0 |
| | Global KL beta | 0 |
| | Free nats | 3 |
| | Bit depth | 5 |
| Dreamer & Dream-MPC | Planning horizon | 15 |
| | Activation function | ReLU / ELU |
| | Model learning rate | 6e-4 |
| | Actor learning rate | 8e-5 |
| Dreamer, Dream-MPC | Critic learning rate | 8e-5 |
| & hybrid Grad-MPC | Adam epsilon | 1e-7 |
| | Grad clip norm | 100 |
| | Discount factor | 0.99 |
| | Horizon discount factor | 0.95 |
| | Action optimization rate | 0.1 |
| | Action noise | 0.2 |
| Dream-MPC | Action reuse coefficient | 0.1 |
| | Candidates | 5 |
| | Optimization iterations | 1 |
| Hybrid Grad-MPC | Action optimization rate | 0.05 |
| | Planning horizon | 1 |
| Hybrid Grad-MPC & PlaNet | Optimization iterations | 10 |
| | Activation function | ReLU |
| | Candidates | 1000 |
| | Elite candidates | 100 |
| PlaNet | Grad clip norm | 1000 |
| | Model learning rate | 1e-3 |
| | Adam epsilon | 1e-4 |
| | Planning horizon | 12 |

— **Appendices continue on next page** —

---

[3]https://github.com/denisyarats/pytorch_sac_ae
[4]https://github.com/yusukeurakami/dreamer-pytorch

# E  SUMMARY OF DIFFERENCES TO HYBRID GRAD-MPC

We summarize the main differences between Dream-MPC and hybrid Grad-MPC (S V et al., 2023) (also referred to as policy + Grad-MPC by the original authors) as follows:

- **Trajectory optimization.** While the general idea of using a policy to initialize gradient-based MPC is shared by both methods, there are important differences. Dream-MPC uses not just a single trajectory but samples few trajectories from the policy and optimizes each trajectory independently. Additionally, rollout and optimization is performed using longer horizons than just a horizon of one, which is used by hybrid Grad-MPC. While these values can be parameterized, they have a significant impact on the behavior and performance of the optimization. For example, using a horizon of one time step leads to a myopic optimization, which is unsuitable for most problems as outlined in Appendix D. Longer rollouts with learned world models are also more challenging due to imperfect models as shown in Appendix D.2.

- **Uncertainty regularization.** We propose to incorporate uncertainty regularization into the MPC objective, which we find to be particularly important for high-dimensional problems.

- **Action reuse.** We further propose to reuse previously optimized actions instead of completely discarding them to reduce the number of optimization iterations and improve computational efficiency.

- **Extensive experiments and thorough ablations.** Grad-MPC (S V et al., 2023) provides only limited experimental results and lacks in-depth implementation details. While it shows that gradient-based MPC with a policy network is promising for two sparse-reward tasks from the DeepMind Control Suite, it does not provide a full evaluation of the method in diverse settings such as different benchmarks, different world models or types of observations, nor does it address high-dimensional problems, efficiency of gradient-based MPC or analyzes why the performance of gradient-based MPC is usually worse, compared to gradient-free methods. In contrast, Dream-MPC offers a comprehensive set of experiments that systematically analyze the performance of our method across a wide range of conditions, providing new insights into its applicability and efficiency to enable further research.

- **Training with gradient-based MPC.** We also evaluate Dream-MPC when enabling gradient-based MPC already during training and not just during inference. In contrast, hybrid Grad-MPC is only evaluated using pretrained Dreamer models. Our results show that our method is also competitive to gradient-free MPC methods such as MPPI in this setting. In contrast, our experiments with hybrid Grad-MPC showed that it prematurely converges due to the horizon of just one time step.

- **Different world models.** We integrate our method into different types of world models, i.e., Dreamer (generative) and TD-MPC2 (implicit, control-centric) to show that our method is not targeted to a specific world model architecture while (hybrid) Grad-MPC only evaluates their method using Dreamer.

- **Implementation.** Furthermore, we were not able to reproduce the results shown in S V et al. (2023) with the given information because it lacks in-depth implementation details and there is no official implementation available. In contrast, we provide implementation details and will open-source our implementation so that future work can replicate and build upon.

