# OpenReview forum: "Dream-MPC: Gradient-Based Model Predictive Control with Latent Imagination"
_ICLR.cc/2026/Conference — Submitted to ICLR 2026_

### Official Review · Reviewer_paQ4 · 2025-10-27

**Soundness:** 2
**Presentation:** 3
**Contribution:** 1
**Rating:** 4
**Confidence:** 4

**Summary:**

This work introduces Dream-MPC, a gradient-based planning framework that integrates a learned policy prior and world model for trajectory optimization. The method builds on TD-MPC2 (Hansen et al., 2024) by replacing the MPPI planner with a gradient-based differentiable planner commonly found in classical receding horizon MPCs. To stabilize optimization, the authors introduce (1) an uncertainty regularization term and (2) amortization of optimization iterations across time by warm-starting from previous solutions. The authors do extensive evaluations on 24 continuous control tasks from different simulation suites and compare their proposed approach against SAC, DreamerV3, TD-MPC2, and BMPC. The authors report that their method can match or outperform gradient-free methods, particularly when initialized from a strong policy prior.

**Strengths:**

1. **Readable and well-organized**: the paper is clearly well written with effective figures and consistent notation with prior works. The motivation for gradient-based MPC is framed coherently within prior literature of world models (Hafner et al., 2020; Hansen et al., 2022).
2. **Extensive empirical evaluation**: the experiment section is broad and well-presented, covering a variety of domains and considered image-based tasks. The baseline methods are well chosen, and the presented metrics from Agarwal et al., 2021, lend credibility to the reported performance.

**Weaknesses:**

1. **Limited novelty**: the main algorithmic structure is largely inherited from Grad-MPC (S V et al., 2023) and TD-MPC2 (Hansen et al., 2024). The paper's only conceptual addition is the uncertainty penalty (Eq. 5) and minor implementation details. These are relatively minor modifications that do not constitute a substantial advancement in the field.
2. **Questionable motivation for the uncertainty term**: the proposed regularization term $u_t = mean(q_{1:M}) std(q_{1:M})$ is problematic. Maximizing $-u_t$ effectively drives $Q(s,a) \rightarrow 0$, which conflicts with the goal of maximizing $Q(s,a)$. Unlike prior methods (e.g., CQL for uncertainty exploration), this term is not theoretically justified for a general RL algorithm.
3. **Unclear experiment setup**: As it stands, I would not feel confident reproducing Section 5 given the details provided. For example, it is not clear to me which parts of this section are in an offline or online RL setting. While the appendices provide detailed hyperparameters, the authors should consider expanding on the experiment setup and moving some details from the appendix into Section 5.
4. **Weak ablations and inconclusive results**: while I was very excited about this section, I found the number of environments limited and the results mixed. The paper asserts that uncertainty regularization and action reuse are beneficial, yet the shown results do not clearly support that. Critically, there is also missing analysis on the importance of warm starting their planning.

**Questions:**

1. **Sample count**: why do you use only 5 samples for planning? To the best of my knowledge, samples are just bound by available memory (and the gradient buffering required). Personally, I have used 100+ samples in similar settings.
2. **Stochastic optimization**: why don't you treat your planning as a stochastic optimization process and choose to optimize each sampled trajectory independently?
3. **Policy initialization**: how well does your method work if you do not warm-start your planning from a feedback policy?

---

> ### Author Response · Authors · 2025-11-19
> **Author response to reviewer paQ4 (1/2)**
>
> We thank the reviewer for their valuable feedback. We address your comments in the following.
>
> ---
>
> **Regarding limited novelty**
>
> Thank you for your comment. We understand that the novelty of our work might seem limited due to its use of existing techniques. However, we believe that many of the most impactful advancements in reinforcement learning, such as Rainbow, EfficientZero, FastTD3 and Dreamer, which also build on existing methods, nevertheless push the boundaries of existing algorithms in important and valuable ways. In our case, Dream-MPC presents a novel approach to gradient-based MPC by introducing a unique combination of a series of carefully made design decisions informed by large-scale experimental results and supported by existing literature. Specifically, to the best of our knowledge, Dream-MPC is the first gradient-based MPC method with a learned world model that outperforms its gradient-free MPC counterpart across a broad range of different tasks from diverse domains, different types of observations and world models, all while using the same set of planning parameters, neither of which is trivial to achieve **as pointed out by reviewer QH6V**.
>
> ---
>
> **Regarding questionable motivation for the uncertainty term**
>
> While uncertainty regularization is often used for offline RL to mitigate value overestimation [1, 2], it can also be interesting for online RL, especially for off-policy algorithms such as TD-MPC2. The world model may still be queried on unseen state-action pairs during planning, i.e., when estimating returns of candidate trajectories. This can lead to extrapolation errors even if we have learned a good value function [3]. Recent work [4] has shown that TD-MPC2 suffers from value overestimation. To address this, we propose to incorporate uncertainty regularization into the MPC objective, which balances estimated returns and (epistemic) model uncertainty when evaluating candidate trajectories.  By minimizing the uncertainty, we aim to choose actions that lead to higher rewards with a high confidence and to avoid actions for which the outcome is highly uncertain, which is important since we need accurate value estimates as an objective for gradient-based optimization. Our motivation is **not** to use uncertainty for exploration. We agree that this is particularly useful in an offline RL setting, but can limit exploration in an online RL setting. However, this is also true for prior methods that predominantly learn an explicitly and consistently conservative value function and/or policy such as the mentioned CQL method [3].
>
> Since this approach requires to specify a coefficient that weighs estimated value relative to uncertainty for a trajectory, we follow the heuristic proposed by [5] and scale the uncertainty regularization term based on the mean value predictions for a latent state. The uncertainty penalty at a latent state for a trajectory is then multiplied by the coefficient $λ_{unc}$ that balances return maximization and uncertainty minimization. We found it sufficient to use a small value, i.e., $λ_{unc}=0.01$, which means that we only apply a small uncertainty regularization. While the authors in [5] also find that small values of the uncertainty regularization coefficient can improve the performance of TD-MPC2 for offline multi-task RL compared to without regularization, our results for online single-task RL show that the performance of TD-MPC2 can decrease with uncertainty regularization. We acknowledge that uncertainty regularization with online (model-based) RL is fairly unexplored, but leave this for future work given that the heuristic with $λ_{unc}=0.01$ leads to meaningful improvements for Dream-MPC across all 24 environments considered in this work.
>
> ---
>
> **Regarding unclear experiment setup**
>
> All methods and baselines used for the experiments are online RL algorithms. There is **no offline RL** considered in this paper at all. We have updated our manuscript and have moved some details from the appendix to the experiment section to make it more concise. We would appreciate if you could please tell us whether this addresses your concerns or specify which details you are exactly looking for.
>
> ---
>
> **Regarding weak ablations and inconclusive results**
>
> We have added _Table 4_, which shows the importance of warm-starting the optimization by using a policy network and have updated our manuscript. _Figure 4a_ clearly shows that the performance of our method (pink line) significantly benefits from uncertainty regularization and reusing previously planned actions when using MPC during training as illustrated by the gap between the learning curve and the ones for without uncertainty regularization (orange line) and without action reuse (purple line). The ablations were only performed on a subset of environments for computational reasons, but cover all three different benchmarks.

---

> > ### Author Response · Authors · 2025-11-19
> > **Author response to reviewer paQ4 (2/2)**
> >
> > **Q:** Why don't you treat your planning as a stochastic optimization process and choose to optimize each sampled trajectory independently?
> >
> > **A:** We indeed optimize each trajectory independently with its corresponding gradient. We have updated _Section 4_ of our manuscript to make this more clear.
> >
> > ---
> >
> > **Q:** Why do you use only 5 samples for planning instead of e.g. 100+ samples?
> >
> > **A:** One of the main motivations of gradient-based MPC is that we don't need to sample hundreds of trajectories, but can more efficiently directly optimize actions by using gradients of an objective function, especially if we have a high-quality proposal. This is particularly interesting for resource-constrained settings such as robotics. While there are prior works such as Grad-CEM [6] or CEM-GD [7], which combine gradient-based and gradient-free MPC and use hundreds of trajectories, these approaches are computationally expensive and still suffer from scalability to higher dimensional problems. Also, as pointed out in the answer of the previous question, we optimize each trajectory independently and thus compute gradients in a batched manner, which also becomes computationally more expensive with an increasing number of trajectories. We found that using five samples performs well for a broad variety of tasks as also illustrated by the parameter sweep in _Figure 5_.
> >
> > ---
> >
> > **Q:** How well does your method work if you do not warm-start your planning from a feedback policy?
> >
> > **A:** We have added results for this ablation in _Section 5.2_ (_Table 4_), which shows that a high-quality initial proposal is in particular important for high-dimensional tasks.
> >
> > ---
> >
> > [1] Fujimoto et al. Off-policy deep reinforcement learning without exploration. ICML 2019.
> >
> > [2] Kumar et al. Stabilizing off-policy q-learning via bootstrapping error reduction. NeurIPS 2019.
> >
> > [3] Feng et al. Finetuning offline world models in the real world. CoRL 2023.
> >
> > [4] Lin et al. TD-M(PC)^2: Improving temporal difference MPC through policy constraint. 2025. https://arxiv.org/abs/2502.03550
> >
> > [5] Hansen et al. TD-MPC2: Scalable, robust world models for continuous control. ICLR 2024.
> >
> > [6] Bharadhwaj et al. Model-predictive control via cross-entropy and gradient-based optimization. L4DC 2020.
> >
> > [7] Huang et al. CEM-GD: Cross-entropy method with gradient descent planner for model-based reinforcement learning. 2021. https://arxiv.org/pdf/2112.07746

---

### Official Review · Reviewer_QH6V · 2025-10-31

**Soundness:** 3
**Presentation:** 4
**Contribution:** 4
**Rating:** 8
**Confidence:** 3

**Summary:**

Model Predictive Control (MPC) is a method that uses a world model (WM) or simulator to plan actions before taking them in a target environment (like the real world).  MPC looks $k$ steps into the future, and finds the trajectory which has the highest reward, then generally only takes the first action.  Finding good actions in the WM is normally either done by sampling from a policy, a gradient free populations based methods, or some hybrid of the 2.

The WM can used to learn a reactive policy, as done in Dreamer.  However, purely reactive policies can have limited generalization.  Gradient free model free optimization methods like Cross Entropy Method (CEM) scale can poorly as you increase the number of dimensions.  In contrast, gradients perform very well in optimizing very high dimensional spaces and thus should be a natural fit for optimizing the trajectories.

Past approaches to gradient based MPC have performed poorly.  The authors introduce a novel methods called Dream-MPC that uses gradient ascent to optimize trajectories that overcomes some of these shortcomings.  They start by sampling a small number of trajectories from the policy, then proceed to optimize the actions using gradient ascent, in each iteration recomputing the latent states from the rolled out actions.  In order to not overfit, uncertainty regularization is added to the gradient target as well.

MPC works by replanning to the end of the full window, but then only taking a first (few) actions.  In later iterations, the old "tails" from the previous trajectories are mixed in with the new samples from the polices, amortizing the optimization iterations over time.

They integrate their gradient based action optimization method into TD-MPC2 and BMPC - two existing model based reinforcement learning algorithms. They go on to show that they significantly outperform existing model based reinforcement learning algorithms on a wide variety of tasks from deepmind control, humanoid bench and meta world.

Finally, they perform ablation studies, demonstrating the importance of warm starting with policy samples,  uncertainty regularization, and even the benefit of using gradient ascent at all, showing each part is important for the final performance.

**Strengths:**

## Originality
While gradient optimization on top of MPC is definitely not a novel idea, the authors combined many pieces to actually get a SOTA system, which in itself is an original contribution.

## Quality
The model is well formulated, and motivated, and the differences from the existing methods in the filed are clear.  Also, the experiments are well run, rigorously defined, and clearly demonstrate the superiority of their method.  The ablations clearly motivate each of their model decisions.  They are careful in how they compare RL performance between the different methods as well.

## Clarity
The problem to solve is clearly laid out, and are good at motivating the approach taken.  It should be possible to replicate the results from the descriptions in the paper.

## Significance
The results clearly show their method is better.  Unlocking gradient based optimization in combination with world models is also likely to result in further research and impacts as other researches seek to improve upon their innovations.

**Weaknesses:**

Nothing specific comes to mind.

**Questions:**

Did you consider using more powerful optimizers like ADAM, etc?  Why did you choose to only use plain gradient ascent?  Alternatively, as the state spaces are small, do natural gradient style methods help the optimization process?

---

> ### Author Response · Authors · 2025-11-19
> **Author response to reviewer QH6V**
>
> We thank the reviewer for their valuable feedback and are glad that you consider our work to be clearly written, well-motivated, addressing an important problem, containing extensive and thorough experiments and ablations, and a meaningful contribution to benefit future research. We address your question in the following.
>
> ---
>
> **Q:** Did you consider using more powerful optimizers like ADAM, etc? Why did you choose to only use plain gradient ascent?
>
> **A:** Yes, we did limited experiments with other optimizers, but in general found the differences to be relatively small. We hypothesize that this is due to the relatively short prediction horizon of TD-MPC2, but effects may be more pronounced for longer horizons. Thus, we believe this is an interesting aspect for future research.

---

> > ### Comment · Reviewer_QH6V · 2025-11-25
> >
> > Thanks for addressing my question. I'll maintain my score.

---

> > > ### Author Response · Authors · 2025-11-27
> > > **Thank you!**
> > >
> > > Dear Reviewer QH6V,
> > >
> > > We thank you again for your review and engagement with our paper and are glad that you like our paper!
> > >
> > > Best,
> > >
> > > Authors of Dream-MPC (Submission 17302)

---

### Official Review · Reviewer_xthc · 2025-11-01

**Soundness:** 2
**Presentation:** 3
**Contribution:** 2
**Rating:** 2
**Confidence:** 3

**Summary:**

The paper introduces Dream-MPC, a gradient-based model-predictive controller that warm-starts each optimisation episode with a **stochastic policy prior, re-uses past optimised actions, and penalises uncertainty**. Empirical results on 24 tasks (DM-Control, HumanoidBench, Meta-World) and four image-based settings show that the method improves the underlying policy and, when coupled with only BMPC, matches or exceeds MPPI.

## Main Experiments

- Offline checkpoints: Replace MPPI in **TD-MPC2/BMPC with Dream-MPC** at test time (1 it × 5 candidates, H = 3).
- Online training: Learn world-model + actor with Dream-MPC planner on four DM-Control tasks; compare against Dreamer, PlaNet, SAC+AE, Grad-MPC, hybrid Grad-MPC.
- Ablations: Remove **action-reuse, uncertainty term, and policy prior**. sweep over horizon/iterations/candidates.
- Runtime: Inference latency on RTX-4090.

**Strengths:**

## Strengths

- Dream-MPC is designed as a **drop-in module** that slots into popular world-model agents (Dreamer-v3, TD-MPC2, BMPC) without altering their training loops. This practical compatibility lowers the barrier to adoption in both research and real-time robotics deployments.

-  The authors include a thorough **gradient-variance and ESNR analysis across horizons**, an insight likely to inform future gradient-based MPC work.

- Low Sensitivity to Candidate Count : An ablation shows that performance saturates with as few as five candidate trajectories, indicating robustness to the primary computational knob and making the method attractive for resource-constrained settings.

**Weaknesses:**

## Weaknesses

- **Generality of the Method**: The paper's framing of the method's effectiveness could be moderated. The empirical results suggest the performance improvements are not uniform. For instance, the gains when integrating with TD-MPC2 appear to be minimal or, in some cases, negative , while the improvement over the **strong BMPC** baseline is modest. This selective efficacy suggests the method is more of a targeted enhancement for specific model-based architectures (like BMPC) rather than a universally applicable solution for all world models. As confirmed in **Figure 3 and 4**.

- Confounded Attribution of Gains: The attribution of performance gains is somewhat confounded by the experimental design. Key parameters, such as learning-rate schedules and candidate budgets (e.g., **5 for Dream-MPC vs. 512 for MPPI**), differ between the proposed method and the baselines. This makes it challenging to isolate the specific contribution of the novel components, such as the uncertainty penalty, from the effects of other hyperparameter choices.

- Incremental novelty: The present work mainly extends Hybrid Grad-MPC with the horizon and adds a small regularizer. Functional and performance overlap is acknowledged in **Table 14 and 15**. Methodological delta over Hybrid Grad-MPC may be modest (longer horizon + variance penalty).

**Questions:**

## Read Weaknesses

- TD-MPC2 integration – Why does substituting Dream-MPC for MPPI yield clear gains in BMPC, yet provide little or no improvement when plugged into TD-MPC2? Could the authors elaborate on where this problem comes from?.

- Uncertainty penalty – Why is **λ_unc fixed to 0.01, Action-reuse coefficient to ρ=0.1** chosen heuristically, no sensitivity study.

Additional Remarks

- Hybrid Grad-MPC equivalence: The manuscript states that “Hybrid Grad-MPC is equivalent to Dream-MPC with a single candidate trajectory and a horizon of one,” yet immediately notes that the results could not be reproduced. I think, Hybrid Grad-MPC treat **candidate count and horizon** as tunable hyper-parameters and evaluate primarily on **sparse-reward tasks**. the current framing may inadvertently misrepresent that work's contribution. A clearer explanation (or removal) of the equivalence claim would avoid confusion.

- Hyper parameter Accessibility: The paper's readability would be improved by moving essential hyper parameters from the appendix into the main text. Key settings, such as the λ_unc coefficient and the planning/optimization parameters, are critical to understanding the method. A **concise table in the main body would allow readers to grasp the method's essential configuration** without needing to frequently reference the appendix, like PPO table listed in Dreamer V3 paper.


Given the method’s limited generality, incremental gains & novelty, unclear attribution of improvements, it does not meet the significance threshold for acceptance.

---

> ### Author Response · Authors · 2025-11-19
> **Author response to reviewer xthc (1/3)**
>
> We thank the reviewer for their valuable feedback. We address your comments in the following.
>
> ---
>
> **Regarding generality of the method**
>
> As shown by our experiments with different types of world models using TD-MPC2 (implicit, control-centric) and Dreamer (generative), our method is applicable to different methods even with the same planning parameters and is not targeted to a specific model-based architecture, but can be integrated into different world models such as Dreamer or TD-MPC2 as you point out. We further want to emphasize that the results in _Figure 3_ and _Figure 4_ do not use BMPC, but TD-MPC2 as a basis, which suffers from a large performance gap between the policy network and MPPI. Although combining gradient-based MPC with the bootstrapping approach of BMPC is an interesting research direction, we leave this for future work because we hypothesize that it might lead to unstable training and premature convergence.
>
> While Dream-MPC can significantly improve the performance of the underlying policy by **144.7%** (IQM) and **43.4%** (mean), it can overall not match the performance of MPPI in this case as you pointed out and as we describe in the paper. Combined with BMPC, Dream-MPC improves the IQM by **26.7%** and the mean by **20.5%** compared to BMPC, outperforming all baselines as also shown in _Figure 2_ and _Table 2_. Thus, we show that a high-quality proposal is required for efficient gradient-based MPC whereas sampling-based MPC methods such as MPPI may be better if we don't have a good initial proposal. We also discuss the reasons for this in more detail in the answer to your question below. We have updated the manuscript to make this more clear.
>
> ---
>
> **Regarding attribution of performance gains**
>
> Since Dream-MPC (gradient-based) and MPPI (gradient-free) are completely different methods, they cannot be used for a direct comparison of design decisions or hyperparameters. For a fair comparison, we use the default hyperparameters for MPPI and a comparable computational budget as illustrated in _Table 3_ (_Table 2_ in the initial submission). We also want to emphasize that the only parameter differences are the ones specific to the planning methods because we do not change any hyperparameters of the underlying base methods. To isolate the specific contributions of the different components such as the uncertainty regularization, we perform ablation studies in _Section 5.2_ and also provide results for MPPI with uncertainty regularization in _Appendix C.3_. _Tables 12-14_ (in the initial submission _Tables 11-13_) show that uncertainty regularization for TD-MPC2 actually decreases performance, especially for HumanoidBench. While uncertainty regularization slightly improves performance for BMPC on DMControl and Meta-World, the results for HumanoidBench are also worse compared to no uncertainty regularization.
>
> ---
>
> **Regarding incremental novelty**
>
> As already described in our paper, we acknowledge that both methods share the same idea, i.e., using a policy network to warm-start gradient-based MPC. However, there are important differences such as the uncertainty regularization and the action reuse, which we have summarized in _Appendix E_. We would also like to emphasize that incremental nature applies to many RL methods such as Rainbow, FastTD3 or Dreamer, which have nevertheless made significant and important contributions, and also for (hybrid) Grad-MPC itself, which can be seen as a combination of prior works such as Grad-CEM [1] and POPLIN [2] or as an incremental version of MBOP [3] as even acknowledged by the Grad-MPC authors themselves. Furthermore, we want to highlight that it is not trivial to actually accomplish a well-working gradient-based MPC method that achieves state-of-the-art-performance as also pointed out by **reviewer QH6V**. We also want to refer to our detailed answer for **reviewer paQ4** to avoid duplicated answers.

---

> > ### Author Response · Authors · 2025-11-19
> > **Author response to reviewer xthc (2/3)**
> >
> > **Q:** Why does substituting Dream-MPC for MPPI yield clear gains in BMPC, yet provide little or no improvement when plugged into TD-MPC2? Could the authors elaborate on where this problem comes from?
> >
> > **A:** When comparing the results in _Tables 9-11_ (_Tables 8-10_ in the initial submission) to _Tables 12-14_ (_Tables 11-13_ in the initial submission), we can see that Dream-MPC also significantly improves the performance of the underlying policy for TD-MPC2. We have added a summarized table below to make the comparison easier. However, Dream-MPC is in general not able to match the performance of MPPI. The main reason for this is that the underlying policy is simply not good enough, especially for high-dimensional tasks such as HumanoidBench, which is indicated by the large gap between the policy only and standard TD-MPC2 using MPPI. While we found that we can further improve the performance for example by increasing the number of optimization iterations, this also increases the computational effort. Thus, we decided to report the results as they are for the default planning parameters to have a fair comparison to MPPI. While we have already discussed this in the appendix of the initial submission, we have emphasized this now also in the main text (_Section 5.1_) of the updated version of the manuscript.
> >
> > | Environment             | TD-MPC2     | TD-MPC2 (policy-only) | Dream-MPC (TD-MPC2)   |
> > |-------------------------|-------------|-----------------------|-----------------------|
> > | **DM Control**          |             |                       |                       |
> > | Acrobot Swingup         | 595 ± 34    | 551 ± 21              | 590 ± 40              |
> > | Cartpole Swingup Sparse | 848 ± 0     | 760 ± 114             | 847 ± 3               |
> > | Fish Swim               | 786 ± 8     | 645 ± 83              | 764 ± 56              |
> > | Hopper Hop              | 493 ± 47    | 383 ± 154             | 307 ± 38              |
> > | Dog Run                 | 358 ± 228   | 89 ± 52               | 115 ± 72              |
> > | Dog Walk                | 933 ± 10    | 298 ± 20              | 389 ± 22              |
> > | Humanoid Run            | 344 ± 60    | 65 ± 2                | 110 ± 10              |
> > | Humanoid Walk           | 899 ± 10    | 142 ± 36              | 338 ± 63              |
> > | **Meta-World**          |             |                       |                       |
> > | Assembly                | 1.0 ± 0.0   | 1.0 ± 0.0             | 1.0 ± 0.0             |
> > | Button Press            | 0.33 ± 0.47 | 0.0 ± 0.0             | 0.33 ± 0.47           |
> > | Disassemble             | 0.67 ± 0.47 | 0.67 ± 0.47           | 0.67 ± 0.47           |
> > | Lever Pull              | 0.0 ± 0.0   | 0.0 ± 0.0             | 0.0 ± 0.0             |
> > | Pick Place Wall         | 1.0 ± 0.0   | 0.0 ± 0.0             | 0.67 ± 0.47           |
> > | Push Back               | 0.67 ± 0.47 | 0.33 ± 0.47           | 0.67 ± 0.47           |
> > | Shelf Place             | 0.67 ± 0.47 | 0.67 ± 0.47           | 1.0 ± 0.0             |
> > | Window Open             | 1.0 ± 0.0   | 0.33 ± 0.47           | 0.67 ± 0.47           |
> > | **HumanoidBench**       |             |                       |                       |
> > | Balance Hard            | 92 ± 12     | 34 ± 3                | 45 ± 10               |
> > | Balance Simple          | 240 ± 37    | 33 ± 16               | 47 ± 14               |
> > | Hurdle                  | 78 ± 24     | 14 ± 3                | 12 ± 1                |
> > | Maze                    | 169 ± 47    | 111 ± 3               | 120 ± 8               |
> > | Reach                   | 5037 ± 1436 | 1558 ± 368            | 2751 ± 444            |
> > | Run                     | 136 ± 110   | 8 ± 4                 | 10 ± 7                |
> > | Slide                   | 237 ± 54    | 14 ± 2                | 16 ± 3                |
> > | Stair                   | 100 ± 18    | 24 ± 8                | 30 ± 6                |
> >
> > ---
> >
> > **Q:** Why is $λ_{unc}$ fixed to 0.01, Action-reuse coefficient to $ρ=0.1$ chosen heuristically?
> >
> > **A:** We have performed a parameter search and have found that these parameters perform empirically well across different benchmarks, observation types (states and images), and world models (TD-MPC2 and Dreamer). Intuitively, small values for both parameters are expected. We have also added a sensitivity analysis of the parameter values in _Section 5.2_ (_Figure 4b_).
> >
> > ---
> >
> > **Q:** Hybrid Grad-MPC equivalence
> >
> > **A:** Dream-MPC provides a general framework for gradient-based MPC from which hybrid Grad-MPC can also be derived. We want to emphasize that parameter choices like the horizon are not minor implementation details, but crucial because optimizing for more than just one time step in the future with learned world models is not trivial due to imperfect models. We have refined the explanation to avoid confusion and hope that we have addressed your concerns. Please let us know if not!

---

> > > ### Author Response · Authors · 2025-11-19
> > > **Author response to reviewer xthc (3/3)**
> > >
> > > **Q:** The paper's readability would be improved by moving essential hyperparameters from the appendix into the main text.
> > >
> > > **A:** While key parameters such as $λ_{unc}$, the horizon $H$ or the number of candidate trajectories $N$ were already mentioned in the main text before, we have updated our manuscript and included the parameters specific for Dream-MPC in the main text (_Table 1_) to make it more concise. Because of space limitations, we initially followed the common practice of listing the detailed parameters in the appendix as also done in the Dreamer-v3 paper.
> > >
> > > ---
> > >
> > > [1] Bharadhwaj et al. Model-predictive control via cross-entropy and gradient-based optimization. L4DC 2020.
> > >
> > > [2] T. Wang and J. Ba. Exploring model-based planning with policy networks. ICLR 2020.
> > >
> > > [3] A. Argenson and G. Dulac-Arnold. Model-based offline planning. 2021. http://arxiv.org/abs/2008.05556

---

> > > > ### Comment · Reviewer_xthc · 2025-11-28
> > > >
> > > > Thank you for the detailed rebuttal and for the additional tables and clarifications. I appreciate the revisions and clarifications.
> > > > My remaining concerns are grouped below:
> > > >
> > > > 1. **Regarding generality of the method** : In my review, I was also referring to performance of Dream MPC with TD-MPC in figure 3 and 4. In these figures, I do not observe a consistent improvement of Dream-MPC with TD-MPC over the baselines. I agree Dream-MPC in conjunction with BMPC show significant gains.
> > > >
> > > >
> > > > 2. **Clarification of Novelty Claims** : While I appreciate the technical merit of the approach, I remain unconvinced that the core idea, using a policy to generate trajectory samples for gradient-based MPC is novel. This idea has been previously explored, for instance, in Hybrid Grad-MPC and related works such as MBOP,  MPC via CE
> > > > and gradient-based optimization and Grad-MPC. Thus, describing the method as “a novel approach that generates few candidate trajectories from a rolled-out policy” may overstate its originality.
> > > >
> > > >  **I recommend, changing this phrase from the abstract and main text and clearly acknowledging the method as an incremental improvement over prior work**. Explicitly citing previous works like Hybrid Grad-MPC and positioning the contribution as an empirical advance would improve merits of paper.
> > > >
> > > > 3. **TD-MPC2 / Dream-MPC comparison (table in the rebuttal)**: The summarized table for TD-MPC2, TD-MPC2 (policy-only) and Dream-MPC (TD-MPC2) actually reinforces my impression that Dream-MPC underperforms TD-MPC2 in the standard setting. Moreover, once standard deviations are taken into account, TD-MPC2 (policy-only) seems to perform on par with Dream-MPC on many tasks, with only a few cases later showing noticeable improvement. As RL Practitioner, I do not see strong evidence of a **substantial and consistent performance gap** in favor of Dream-MPC.
> > > >
> > > >
> > > > 4. **Hyper parameter placement** : Thank you for moving key hyper parameters into the main text (Table 1) and conducting additional experiments on hyper parameters. **This is a helpful improvement for reproducibility and reader clarity**.
> > > >
> > > >
> > > > Overall, I would suggest to make substantial changes to manuscript before publication and highlight limitations (my review and other reviews) of the Dream-MPC method. Addressing these items should considerably strengthen the paper’s transparency and impact.

---

> > > > > ### Author Response · Authors · 2025-11-28
> > > > > **Author response to reviewer xthc**
> > > > >
> > > > > Dear Reviewer xthc,
> > > > >
> > > > > Thank you for acknowledging our response! We are glad to hear that our response has addressed most of your concerns. We address your remaining concerns below.
> > > > >
> > > > > ---
> > > > >
> > > > > **Regarding generality of the method**
> > > > >
> > > > > _Figure 3_ aims to study whether we can also already use gradient-based MPC during training as an alternative to MPPI. We are to the best of our knowledge the first to show that this is possible without premature convergence. Dream-MPC with TD-MPC2 can improve performance compared to TD-MPC2 with MPPI for lower dimensional problems. While the performance for high-dimensional problems such as _Dog Run_ or _Reach_ is worse compared to TD-MPC2 with MPPI, it does not completely fail to learn. _Figure 4_ shows that this can be particularly attributed to the uncertainty regularization and action reuse. We have also described this in _Section 5.1_ and _Section 5.2_. While we agree that future research should aim to further improve the performance in this setting, we believe that future work benefits from our insights and contributions.
> > > > >
> > > > > ---
> > > > >
> > > > > **Regarding clarification of novelty claims**
> > > > >
> > > > > We have acknowledged previous works on policy-guided MPC and gradient-based MPC already in our initial submission, which also explicitly cites the mentioned Grad-MPC paper in the _Related Work_ section (_Section 2_). While there are previous works, which either optimize hundreds of trajectories rolled out from a policy network by using gradient-free methods such as CEM or MPPI (e.g. TD-MPC2), or perform gradient-based MPC with hundreds of trajectories from a Gaussian distribution or a single trajectory rolled out from a policy network (Grad-MPC), the combination of rolling out only few trajectories from a policy network **and** optimizing each trajectory by gradient ascent is novel, particularly with our proposed uncertainty regularization and action reuse. In our previous response we have also highlighted that we have added a **summary of the main differences between Dream-MPC and hybrid Grad-MPC in _Appendix E_** (which we also refer to in _Section 2_) to clarify our contributions. We further want to stress that the description of our method in the abstract is emphasizing exactly this unique combination of policy-guided MPC and gradient-based MPC since the complete sentence is "We propose Dream-MPC, a novel approach that generates few candidate trajectories from a rolled-out policy **and optimizes each trajectory by gradient ascent using a learned world model**". The proposed uncertainty regularization and action reuse are described in the next sentence. We have revised our claims to make our contributions more clear, e.g. _lines 135 -- 142_.
> > > > >
> > > > > ---
> > > > >
> > > > > **Regarding TD-MPC2 / Dream-MPC comparison (table in the rebuttal)**
> > > > >
> > > > > As we stated before, **Dream-MPC (TD-MPC2) improves the overall mean performance by 43.4% compared to the underlying policy**. We agree that performance improvements of Dream-MPC (TD-MPC2) compared to the underlying policy vary across different environments. As we acknowledge in our previous answer, Dream-MPC (TD-MPC2) is for high-dimensional problems such as humanoid robots not able to match the performance of TD-MPC2 with MPPI because it is limited by the quality of the underlying policy. For lower-dimensional problems such as _Acrobot Swingup_, _Cartpole Swingup Sparse_, _Fish Swim_ or _Meta-World_, Dream-MPC (TD-MPC2) can match the performance of TD-MPC2 with MPPI. We emphasize this, i.e., that gradient-based MPC requires a high-quality initial proposal in _Section 5.1_ (lines 356 -- 359). As you agree, Dream-MPC in conjunction with BMPC, which has a stronger underlying policy, shows significant gains compared to MPPI. While we found that we can further improve the performance of Dream-MPC with TD-MPC2 as a basis for example by increasing the number of optimization iterations, this also increases the computational effort. Thus, we decided to report the results as they are to have a fair comparison to MPPI in terms of computational time (cf. inference times in _Table 3_). We believe that our manuscript realistically and transparently demonstrates the potential and limitations of gradient-based MPC, thereby contributing to future research in this field.
> > > > >
> > > > > ---
> > > > >
> > > > > We hope that our responses have addressed the remaining reviewer's concerns. Please let us know if not! Otherwise, we kindly ask the reviewer to consider raising their score accordingly.

---

### Official Review · Reviewer_1niH · 2025-11-02

**Soundness:** 4
**Presentation:** 3
**Contribution:** 3
**Rating:** 6
**Confidence:** 4

**Summary:**

The paper proposes Dream-MPC, a novel gradient-based Model Predictive Control (MPC) method, aiming to address the issues of high computational cost and low efficiency in high-dimensional tasks of traditional hybrid MPC (which relies on gradient-free optimization), as well as the problems of traditional gradient-based methods being prone to falling into local optima and gradient anomalies. It generates a small number of candidate trajectories (5 by default) from a policy network, optimizes them using gradient ascent with a learned world model, incorporates uncertainty regularization (estimating uncertainty based on an ensemble of Q-functions and penalizing trajectories with high uncertainty in the objective), and realizes the amortization of optimization iterations by reusing previously optimized actions. Validated on continuous control tasks, it improves the performance of the underlying policy (e.g., achieving the best average score when based on BMPC) and features fast inference speed. It can be flexibly integrated into frameworks such as TD-MPC2, BMPC, and Dreamer.

**Strengths:**

The experiments in the paper are sufficiently comprehensive and yield rich results, illustrating the effectiveness and limitations of the proposed Dream-MPC method from multiple perspectives, with a clear overall presentation. Grad-MPC’s effectiveness is validated and its applicable scenarios are analyzed through targeted experiments.

**Weaknesses:**

In the context of the document, the research exhibits clear merits: its experimental scope is extensive, encompassing a diverse set of continuous control tasks across multiple domains, and the results effectively validate the proposed Dream-MPC method’s performance advantages. At the same time, there are subtle areas where the presentation could be further polished. For one, the distinctions between Dream-MPC and configurations like "policy + Grad-MPC" are not fully and explicitly articulated, which may leave room for readers to desire more clarity on this comparative aspect. Additionally, while the abstract and conclusion position Dream-MPC as a solution to the limitations of earlier gradient-based MPC methods, the detailed content explaining how it specifically addresses these prior shortcomings is mostly contained within the appendix rather than the main text—where the narrative focus is more heavily placed on showcasing Dream-MPC as a new high-performing approach through comparisons with existing baselines. Refining these aspects slightly could help strengthen the document’s coherence in linking its innovative points to the gaps of previous methods.

**Questions:**

- What are the differences between the method in this paper and the "Policy + Grad-MPC" approach described in Section 6 of the paper Gradient-based Planning with World Models (https://arxiv.org/pdf/2312.17227)? The latter mentions that "The Policy+Grad-MPC method operates in a manner similar to the Grad-MPC method explained in previous sections. However, in this approach, trajectories are initialized from the output of the policy network." Do the innovations of this paper overlap with this description?

- If the method in this paper is an improvement over Grad-MPC, it is recommended to add the definition and introduction of Grad-MPC in the "PRELIMINARIES" section. Then, highlight the differences between the proposed method and the original Grad-MPC in the "METHOD" chapter, or add a section in the main text to summarize the differences from Policy+Grad-MPC.

- Figure 2: Aggregate performance metrics, why are the results of Dream-MPC (TD-MPC2) not included here, but only placed in the appendix? Overall, a high-quality policy is crucial for the method in this paper; it is recommended to emphasize this point in the main text rather than in the appendix.

- Lines 416 to 424 state: "We further evaluate the performance of fully trained TD-MPC2 agents with gradient-based MPC when varying the number of candidates, the number of optimization iterations, and the planning horizon". However, the BMPC version is shown in the result figure. Is there a wrong image attached or a typo?

- In the CONCLUSION, it is stated that "Our empirical evaluation shows that Dream-MPC can not only outperform the baselines, but is also more robust to hyperparameters and faster compared to previously proposed gradient-based MPC methods." Could you please inform which table or figure illustrates the claim that Dream-MPC is "more robust to hyperparameters compared to previously proposed gradient-based MPC methods"? Is it Section D.2? If so, what are the differences between Grad-MPC and Dream-MPC in the experiments conducted here, and could you please explain them?

---

> ### Author Response · Authors · 2025-11-19
> **Author response to reviewer 1niH**
>
> We thank the reviewer for their valuable feedback. We address your comments in the following.
>
> ---
>
> **Regarding refining presentation**
>
> Thank you for your suggestions on further refining the paper to make it more clear and concise. We have updated our manuscript to make it more coherent and clarify the differences between Dream-MPC and hybrid Grad-MPC. We hope that we have addressed the reviewer's concerns. Please let us know if not!
>
> ---
>
> **Q:** What are the differences between Dream-MPC and hybrid Grad-MPC (policy + Grad-MPC)?
>
> **A:** While both methods share the general idea of warm-starting gradient-based MPC by using a policy network, there are significant differences such as the uncertainty regularization and the reuse of optimized actions. We have added _Appendix E_, which explains the main differences between both methods, and we have also added a brief summary in the main text (_Section 2_).
>
> ---
>
> **Q:** Why are the results of Dream-MPC (TD-MPC2) not included in the aggregated performance metrics (Figure 2), but only placed in the appendix?
>
> **A:** We have initially not included it to keep the figure concise, but have now added the results to _Figure 2_ and have emphasized the importance of a high-quality proposal in the main text. Additionally, we have also added an ablation for not using the policy for warm-starting MPC in _Section 5.2_ (_Table 4_).
>
> ---
>
> **Q:** Is there a typo in the description (lines 416-424) of Figure 5 (planning parameter sweep)?
>
> **A:** Yes, this is indeed a typo because we use BMPC as described in the figure caption and legend. We have corrected it in the revised version of our manuscript. Thanks for pointing this out!
>
> ---
>
> **Q:** Could you please inform which table or figure illustrates the claim that Dream-MPC is "more robust to hyperparameters compared to previously proposed gradient-based MPC methods"? Is it Section D.2? If so, what are the differences between Grad-MPC and Dream-MPC in the experiments conducted here, and could you please explain them?
>
> **A:** You are correct that _Section D.2_ is the most relevant one for analyzing the robustness. By analyzing the planner gradients, we found that with increasing horizons the magnitude and variance of the gradients increases for both methods and also when using the ground truth dynamics, i.e., the simulator. Specifically, the gradient variance explodes for longer horizons when using Grad-MPC with a learned dynamics model. In contrast, the variance for Dream-MPC also increases, but remains bounded. We believe that this behavior strongly depends on the smoothness of the optimization landscape, which also aligns with our finding that the Expected Signal-to-Nosie Ratio (ESNR) remains higher and more stable for Dream-MPC compared to Grad-MPC as well as with the findings in [1] who find that a higher ESNR corresponds to more useful gradients.
>
> ---
>
> [1] Georgiev, et al. PWM: Policy learning with multi-task world models. ICLR 2025.

---

### Author Response · Authors · 2025-11-19
**General response and appreciation**

We thank all reviewers for their thoughtful comments. We really appreciate your feedback and are glad that you find our paper clearly written and well-presented (**1niH**, **QH6V**, **paQ4**), the experiments and ablations extensive and thorough (**1niH**, **QH6V**, **paQ4**), that our work addresses an important problem (**xthc**), and provides meaningful insights into gradient-based planning to benefit further research (**xthc**, **QH6V**). We have revised our manuscript based on your feedback and provide a list of changes below. We have also responded to your individual comments.

**Summary of revisions:** We summarize changes to our manuscript below; these changes have also been highlighted (green) in the new version.
* We have **refined the paper to make explanations of our experiments and ablations more concise** (_Section 5.1_ and _Section 5.2_) ([1niH](https://openreview.net/forum?id=uhoCh3pViS&noteId=FGYBxn04Ss), [paQ4](https://openreview.net/forum?id=uhoCh3pViS&noteId=YlPibdBw17)),
* have **moved information** such as the Dream-MPC parameters **from the appendix to the main paper** (_Table 1_) ([xthc](https://openreview.net/forum?id=uhoCh3pViS&noteId=s0O9W6vejE)),
* included an **ablation study for the initial proposal to show the importance of warm-starting by using a policy prior** (_Table 4_) ([paQ4](https://openreview.net/forum?id=uhoCh3pViS&noteId=YlPibdBw17)),
* added a **sensitivity analysis of the uncertainty regularization and action reuse coefficients** (_Figure 4b_) ([xthc](https://openreview.net/forum?id=uhoCh3pViS&noteId=s0O9W6vejE)), and
* added an **overview over the main differences between Dream-MPC and hybrid Grad-MPC** (_Appendix E_) ([1niH](https://openreview.net/forum?id=uhoCh3pViS&noteId=FGYBxn04Ss), [xthc](https://openreview.net/forum?id=uhoCh3pViS&noteId=s0O9W6vejE)).

Again, we thank the reviewers for their constructive feedback. We believe that all comments have been addressed in this revision, but will be happy to address any further comments from reviewers.

Best,

Authors of Dream-MPC (submission 17302)

---

### Meta-Review · Area_Chair_GAf3 · 2026-01-06

**Summary:**

This paper introduces Dream-MPC, a gradient-based MPC variant that combines a policy prior, uncertainty regularization, and action reuse. While the added ablation studies are useful in clarifying the role of these design choices, the overall contribution is largely incremental relative to prior work on policy-guided and hybrid gradient-based MPC. Empirically, the method’s performance depends strongly on the quality of the prior policy. Although this dependence is a valuable insight, it also casts doubt on the robustness and general applicability of the approach. Given the heuristic nature of the regularization and the absence of new conceptual or theoretical advances, the paper does not meet the significance bar for acceptance.

**Reviewer Concerns:**

The rebuttal addressed several clarity and presentation issues, including clearer differentiation from prior work, improved hyperparameter reporting, and additional ablations highlighting the roles of warm-starting, uncertainty regularization, and action reuse.

**Reviewer Scores:**

I speculate the negative reviewers will remain negative, even though they might improve the scores.

---

### Decision · Program_Chairs · 2026-01-26

Reject